# The thioredoxin system determines CHK1 inhibitor sensitivity via redox-mediated regulation of ribonucleotide reductase activity

Chandra Bhushan Prasad [1], Adrian Oo[2], Yujie Liu[1], Zhaojun Qiu[1], Yaogang Zhong[1,3], Na Li[1], Deepika Singh[1], Xiwen Xin[4], Young-Jae Cho[2], Zaibo Li [5], Xiaoli Zhang[6], Chunhong Yan[7], Qingfei Zheng [1,3], Qi-En Wang [1], Deliang Guo [1,3], Baek Kim [2] & Junran Zhang [1,3,8] ✉

Checkpoint kinase 1 (CHK1) is critical for cell survival under replication stress (RS). CHK1 inhibitors (CHK1i's) in combination with chemotherapy have shown promising results in preclinical studies but have displayed minimal efficacy with substantial toxicity in clinical trials. To explore combinatorial strategies that can overcome these limitations, we perform an unbiased high-throughput screen in a non-small cell lung cancer (NSCLC) cell line and identify thioredoxin1 (Trx1), a major component of the mammalian antioxidant-system, as a determinant of CHK1i sensitivity. We establish a role for redox recycling of RRM1, the larger subunit of ribonucleotide reductase (RNR), and a depletion of the deoxynucleotide pool in this Trx1-mediated CHK1i sensitivity. Further, the TrxR inhibitor auranofin, an approved anti-rheumatoid arthritis drug, shows a synergistic interaction with CHK1i via interruption of the deoxynucleotide pool. Together, we show a pharmacological combination to treat NSCLC that relies on a redox regulatory link between the Trx system and mammalian RNR activity.

Lung cancer is one of the most frequently diagnosed cancers. Despite advances in targeted therapy and immunotherapy, lung cancer is still the leading cause of cancer-related deaths worldwide. Thus, the improved treatment approaches are urgently needed. The replication stress (RS) response protein ataxia telangiectasia and Rad3-related protein (ATR) and its main downstream factor, checkpoint kinase 1 (CHK1), play important roles in cell survival under RS[1]. The inhibitors targeting CHK1 (CHK1i's) have been shown to be a powerful strategy for treating solid tumors and hematological malignancies in preclinical studies[2,3]. However, in most clinical trials, including those for treating non-small cell lung cancer (NSCLC), which accounts for 85% of all lung cancer cases, CHKi's have failed to achieve their primary endpoints and have shown cumulative tissue toxicities in normal tissues[4–8]. These findings significantly limit the clinical benefit of these agents. Thus,

[1]Department of Radiation Oncology, James Cancer Hospital and Richard J. Solove Research Institute, The Ohio State University, Columbus, OH 43210, USA. [2]Center for ViroScience and Cure, Department of Pediatrics, School of Medicine, Emory University, Atlanta, GA 30322, USA. [3]The Comprehensive Cancer Center, Center for Cancer Metabolism, The Ohio State University, Columbus, OH 43210, USA. [4]The Ohio State University, Columbus, OH 43210, USA. [5]Department of Pathology, The Ohio State University Wexner Medical Center, College of Medicine, Columbus, OH 43210, USA. [6]Department of Biomedical Informatics, Wexner Medical Center, College of Medicine, The Ohio State University, Columbus, OH 43210, USA. [7]Georgia Cancer Center, Augusta University, Augusta, GA 30912, USA. [8]The Comprehensive Cancer Center, Pelotonia Institute for Immuno-Oncology, The Ohio State University, Columbus, OH 43210, USA. ✉e-mail: Junran.zhang@osumc.edu

identifying combinatorial strategies that can enhance the sensitivity of tumor cells to CHKi's, while limiting their toxicities, might be the key to improve the safety and efficacy of these compounds[3].

The side chain of a proteinogenic cysteine residue contains a terminal thiol (−SH) functional group. The sulfur atom at the core of the thiol is electron rich and its d-orbitals allow for multiple oxidation states[9–12]. Reversible oxidation of thiols by forming the disulfide bond controls protein structure and biological functions. Thiol-based mammalian thioredoxin (Trx) and Grx-GSH systems are important antioxidant systems and maintain a proper balance of protein dithiol (reduced)−disulfide (oxidized) modifications through its disulfide reductase activity[12,13]. The Trx system is comprised of reduced nicotinamide adenine dinucleotide phosphate (NADPH), Trx and thioredoxin reductase (TrxR). Human cells have two members of the Trx family genes; namely, cytosolic Trx1 (encoded by the *TXN* gene) and mitochondrial Trx2 (encoded by the *TXN2* gene)[14]. The Trx system actively removes reactive oxygen species (ROS) and nitrogen species via redox regulation of peroxiredoxins. In addition, the Trx system is also critical for regulating transcription, apoptosis, cell growth and DNA synthesis via redox regulation of ribonucleotide reductase (RNR). However, the regulation of RNR by the Trx system is based on results from *E. coli* and/or yeast. The studies with mammalian cells are limited to the biochemical assays determining RNR activity regeneration by the Trx system, while RNR redox status in vivo has not been investigated[12,15].

RNR promotes the reduction of ribonucleotides to their corresponding deoxyribonucleotides, providing a balanced supply of precursors for DNA synthesis and repair[16]. There are three known human RNR subunits, which are RRM1, RRM2 and p53R2. Based on the pathways of radical initiation and requirements of metal cofactors, the RNRs have been divided into three classes[17], one of which is Class Ia RNR that is found in all types of eukaryotes (including human, mouse and yeast), several viruses, a few prokaryotes (including *Escherichia coli*) and some bacteriophages. Using *E. coli* Class Ia RNR as a prototype model, it has been proposed that the small subunit R2 (equivalent of human RRM2) contains a diiron-tyrosyl radical cofactor and operates as a radical chain initiator by generating a thiyl radical on cysteine Cys[439] in the active site of R1 (equivalent of human RRM1) via a long-range radical transfer pathway[18–20]. Then, the thiyl radical of Cys[439] has been proposed to abstract the 3′ hydrogen atom from nucleoside diphosphate (NDP) and initiate its reduction[21]. As the active site cleft of the R1 subunit is not wide enough to permit the direct reduction by the external redox systems, the reduction of active site disulfide (Cys225 and Cys462) is carried out by a pair of shuttle cysteine residues in the C-terminal mobile tail of R1 subunit (Cys754 and Cys759) and then the resulting disulfide bond in the C-terminal tail of R1 is then reduced by external redox such as Trx or GSH system to continue the next catalytic cycle[17,22]. A similar mechanism occurs in yeast but the Trx system is more important for Grx[23]. In vitro mutagenesis and kinetic studies support a critical role for the C-terminal cysteine pair of R1 in regeneration of the active site[12,17]. Despite the extensive studies in bacteria and yeast, RRM1 redox regulation by the Trx system at the cellular level in human cells has not been reported. Although the role of the Trx system in fueling DNA synthesis during T-cell metabolic reprogramming was suggested via untargeted metabolomics analysis and pathway enrichment analysis under stimulated conditions, this study did not explore the regulatory role of the Trx system in RNR activity, especially with regards to RNR (RRM1)[24].

Active mammalian RNR consist of two homodimers of RRM1 and RRM2 each. In undisturbed cells, the RNR enzyme concentration and composition differ during the cell cycle. The level of RRM1 remains largely unchanged; however, RRM2 increases during the S phase and gets degraded while entering mitosis[25–27]. A recent study identified acetylation of RRM2 as a regulatory mechanism governing RNR activity

to promote cell growth[28]. In response to RS, RRM2 is transcriptionally increased by E2F1 via the ATR-CHK1 pathway[28]. Additionally, RRM1 and RRM2 relocate from the cytoplasm to the nucleus after genotoxic stress induced by UV irradiation[26]. Thus, RNR activity is a highly regulated process in the presence and absence of exogenous DNA damage.

Here, we perform an unbiased high-throughput Decode Pooled shRNA library screen of the NSCLC cell line H1299 to identify the determinants of CHK1i sensitivity. Our screening identifies the *TXN* gene as one of the top hits. Further validation of the screening data show that the depletion of Trx1 or TrxR1 is synthetically lethal with CHK1i treatment due to the interruption of RNR activity in which both RRM1 and RRM2 are involved. Trx1 or TrxR1 depletion leads to an increase in RS because of an accumulation of oxidized RRM1 that results in defective dNTP biosynthesis. In parallel, activation of the E2F1-RRM2 pathway under states of increased RS is significantly restricted by CHK inhibition. Thus, limiting tyrosyl radical transfer activity of RRM2 interrupts long-range radical transfer in RRM1, which further halts dNTP production. Interestingly, we do not observe association of Trx system impairment-induced ROS with the RRM1 redox cycling or its effect on RS nor with the synthetic lethality between Trx system interruption and CHK1i. Auranofin (AUR), a small molecule inhibitor targeting TrxR that has been used clinically to treat rheumatoid arthritis upregulates RRM1 oxidation and shows synthetic lethality with CHK1i due to a depleted dNTP pool. Together, our results reveal that the Trx system serves as a regulator of RRM1 redox recycling in human cells and its inhibition increases RS due to a depleted dNTP pool that renders the cells sensitive to CHK1 inhibition.

## Results
### Trx1 is a determining factor of CHK1i sensitivity

To identify factors that contribute to CHK1i sensitivity, we conducted a high-throughput screen using Decode Pooled Lentiviral shRNA library in the NSCLC cell line H1299 using LY2603618, a first generation specific CHK1i (Fig. 1a). Briefly, cells were transduced with an shRNA library consisting of 95,700 lentiviral shRNAs that targets 18,205 unique human protein-coding genes. shRNA abundance was determined by next-generation sequencing and was compared with untreated cells using model-based analysis of genome-wide shRNA gene knockdown (KD). The effects of the shRNAs were quantified as standardized Z scores (Fig. 1b). Notably, the *TXN* gene, which encodes for Trx1, was identified as one of the top hits as five out of five shRNAs show sensitivity to CHK1i (Fig. 1b). We then focused on both Trx1 and its reductant TrxR1 because of its critical role in Trx1 regulation. Scrutiny of mRNA expression from The Cancer Atlas Genome (TCGA) and ONCOMINE datasets confirmed significant elevation of Trx1 and TrxR1 in NSCLC subsets lung adenocarcinoma (LUAD) and lung squamous cell carcinoma (LUSC) (Fig. 1c and Supplementary Fig. S1a, b). Further data analysis revealed widespread overexpression of Trx1 and TrxR1 in LUAD ($n = 560$; Trx1 $P < 0.00001$; TrxR1 $P < 0.00001$) and LUSC ($n = 546$; Trx1 $P < 0.00001$; TrxR1 $P < 0.00001$). The expression of both Trx1 and TrxR1 in more than 50% of the patients with NSCLC are higher than the 75% quantile expression level in the normal samples (Supplementary Fig. S1c). TCGA data sets also revealed that expression of Trx1 or TrxR1 significantly increases with advancement of the disease (Supplementary Fig. S1d). Trx1 or TrxR1 protein expression was also examined by immunohistochemistry (IHC) in a cohort of tissue microarray (TMA) with their matched adjacent normal tissue (ANT). The median immunoreactive score (IRS) was significantly higher for the NSCLC than ANTs (Fig. 1d). It was found that 88% of the tumor samples were Trx1-positive compared with only 8% of the ANTs and 80% of the sample were positive for TrxR1 positive, compared with only 24% of the ANTs (Fig. 1e). Representative images of NSCLC and

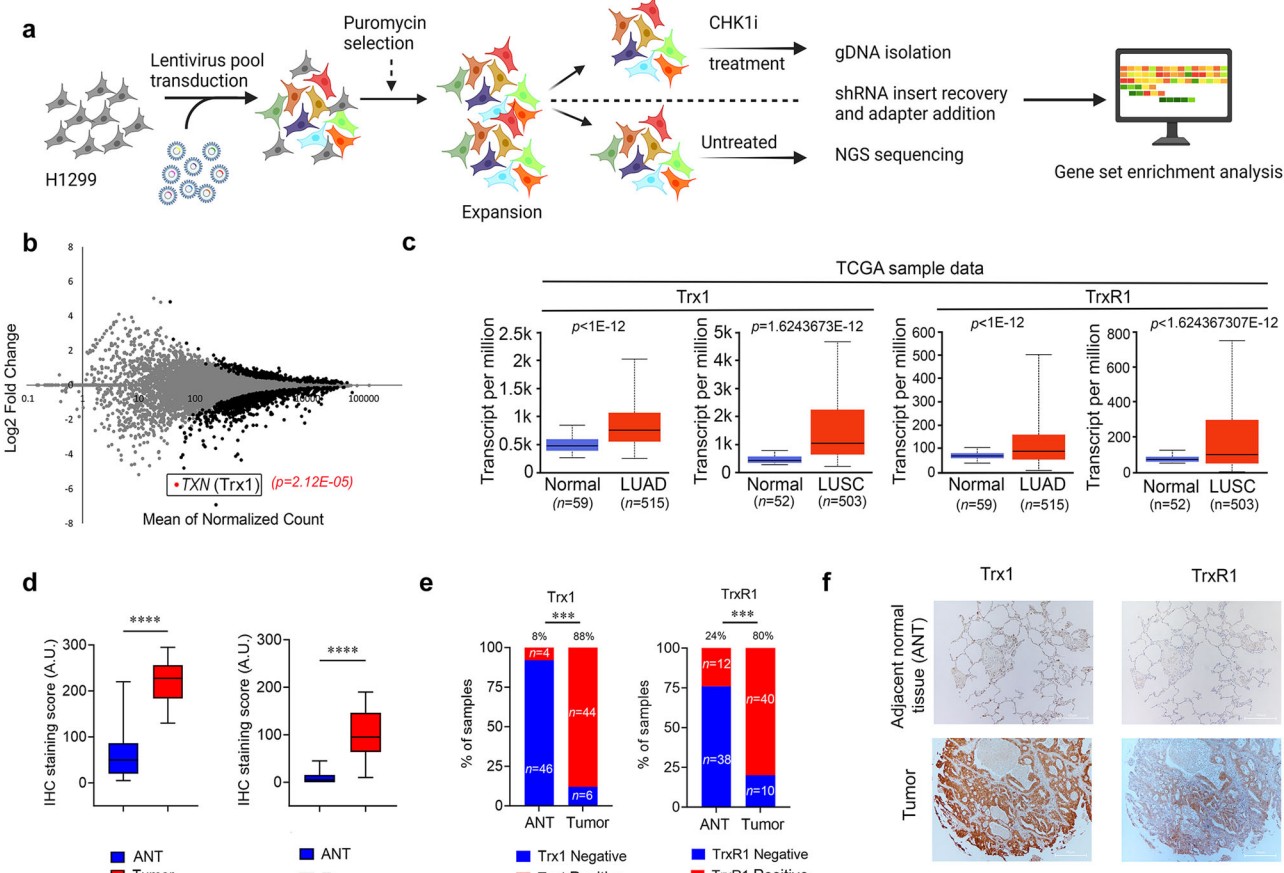

**Fig. 1 | Genome-wide Decode Pooled shRNA library screening identifies Trx1 as a determining factor of CHK1 inhibitor sensitivity. a** Schematic diagram (*Created with BioRender.com released under a Creative Commons Attribution-NonCommercial-NoDerivs 4.0 International license*) illustrating the screening workflow for genome-wide loss-of-function screening. gDNA was isolated from the reference and experimental populations of transduced H1299 cells. CHK1i, CHK1 inhibitor (LY2603618). **b** A volcano plot showing the identified target genes from the screen in **a**. The colored dots are significant genes that have a fold change >1.5 and any point not gray is significant *P* < 0.05. TRX1 is indicated as one of the top candidates from the screen. **c** The analysis of TCGA data sets showing the transcript expression of Trx1 and TrxR1 NSCLC subsets (LUSC & LUAD) compared to normal tissue. Top and bottom of the box indicates the 75th and 25th percentile, respectively. The whisker represents 1.5 times the interquartile range from the box; *p*-values determined by a two-tailed Mann–Whitney test. **d** The median Trx1 and

TrxR1 IHC score (IRS) of the NSCLC tumors (*n* = 50) was significantly higher than that of matched adjacent normal tissue (ANT) (*n* = 50) Top and bottom of the box indicates the 75th and 25th percentile, respectively. The whisker represents 1.5 times the interquartile range from the box; *p*-values determined by a two-tailed Mann–Whitney test. IRS score was determined from the staining intensity (SI) and percentage of positive cells. **e** Bar graph of samples showing positive and negative staining for Trx1 and TrxR1 in tumors and ANT. **f** Representative IHC images of NSCLC tumors (*n* = 50) with matched ANT (*n* = 50); Scale bar, 100 μm. Tissue staining for Trx1 and TrxR1 was performed once on tissue micro array (TMA) containing tumors (*n* = 50 individual tissue) with matched ANT (*n* = 50 individual tissue). [Statistical information: Box plots (**c** and **d**) are represented as median value ± SD. *P* value was determined by a two-tailed Mann–Whitney test.; ***$p \leq 0.0001$; ****$p < 0.0001$].

matched ANT are shown (Fig. 1f). Additionally, higher expression of Trx1 and TrxR1 in NSCLC subsets were positively associated with disease advancement, leading to poor overall survival of patients with LUAD (Supplementary Fig. S1e). Collectively, our results and the data set analysis corroborate previous reports that annotated the association of elevated Trx1 and TrxR1 expression with poor prognosis and survival in different cancers[29–33].

Interestingly, in contrast to a frequent upregulation of the Trx system, defects in Grx-GSH system components compromises GSH homeostasis in NSCLC[34]. Several findings indicate the redundancy between cytosolic Trx1 and the Grx-GSH system, which act as backup systems for each other[34]. The Grx-GSH system and its regulators comprises many components, and a defect in any of them could result in impaired system function. Typically, the sensitivity to Trx1/TrxR1 inhibition is employed as an indicator of Grx-GSH deficiency[34]. Given that NSCLC H1299 cells carry a defective Grx-GSH system[34], our study focuses on the impact of Trx1 and TrxR1 inhibition in NSCLC cells with a defective Grx-GSH system.

## A synergistic interaction occurs between Trx1/TrxR1 KD and CHK1 inhibition

To validate our screening results, we first depleted Trx1 or TrxR1 in two NSCLC cell lines (H1299 and Calu-6) (Fig. 2a), both of which are reported to have a deficient Grx-GSH system, to determine the effect of CHK1i treatment. By cellular toxicity results we found that the Trx1 or TrxR1 depletion increases CHK1i sensitivity, leading to a marked increase in toxicity and cell death in a dose-dependent manner (Fig. 2b). Similarly, treatment with CHK1i for 24 h significantly attenuated the proliferation of Trx1- or TrxR1-depleted cells, compared to scrambled control (shCON) (Fig. 2c, d). Of note, depletion of Trx1 or TrxR1 also magnified ATR inhibitor (ATRi) sensitivity, resulting in cessation of cell proliferation (Supplementary Fig. S2a, b). Additionally, we performed the apoptosis assay in vitro using Annexin V and propidium iodide (PI) staining of NSCLC cells treated with a CHK1i followed by Trx1 or TrxR1KD and found a higher occurrence of apoptosis in the Trx1 or TrxR1 KD cells treated with the CHK1i compared to either condition alone (Supplementary Fig. S3a, b, FACS

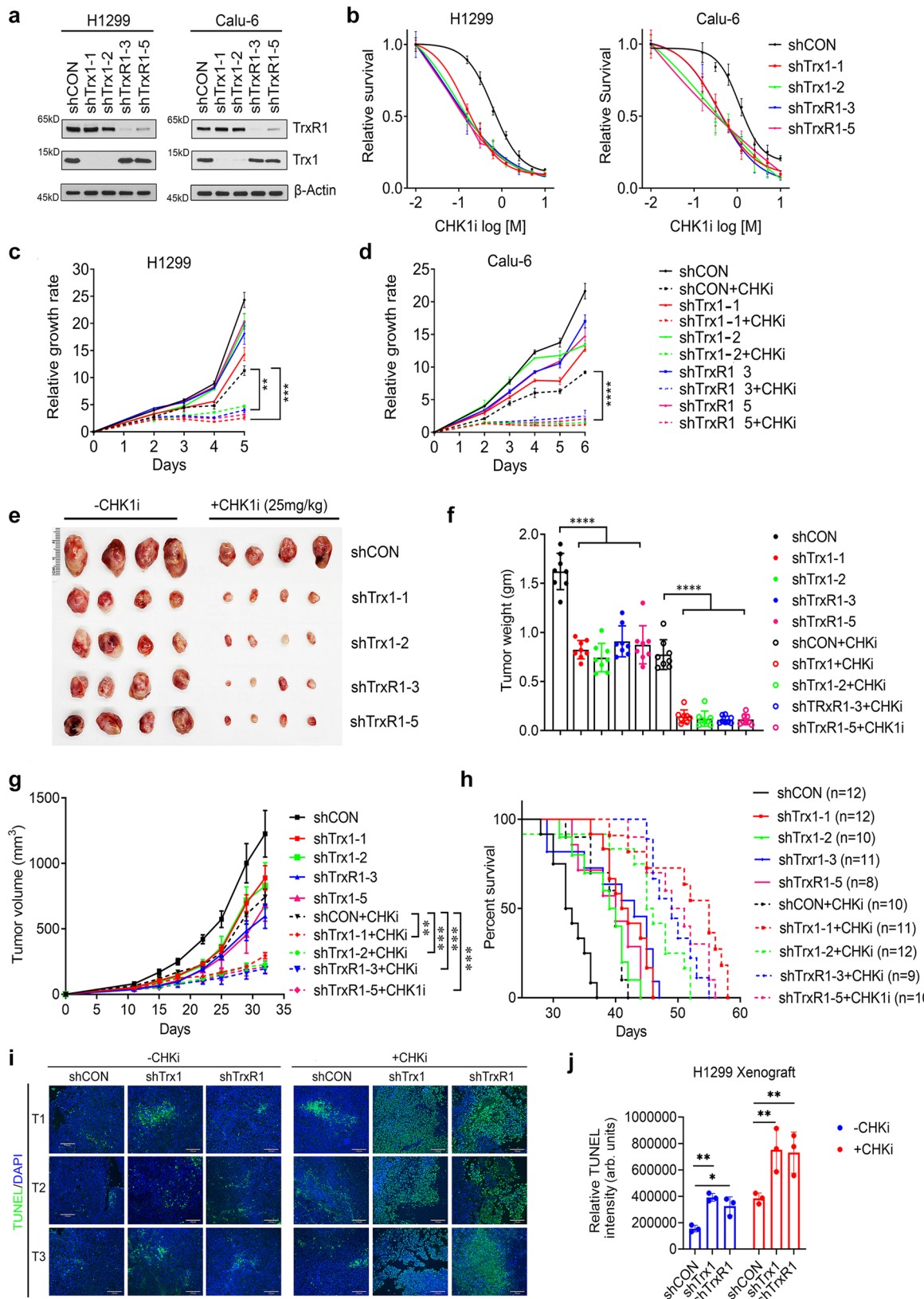

gating strategies are presented in Supplementary Fig. S17). Consistent with the increased apoptosis, a higher cleaved PARP1 in H1299 cells treated with the CHK1i followed by Trx1 or TrxR1 KD was also observed (Supplementary Fig. S3c).

To verify our in vitro results, we conducted an in vivo assay using a xenograft model. We started CHK1i treatment after the tumor reached at least a size of 100 mm³ (Supplementary Fig. S4a). Animals bearing xenograft tumors derived from Trx1- or TrxR1-depleted cells showed a significant decrease in tumor size and tumor weight following CHK1i treatment (Fig. 2e, f), leading to significant inhibition of tumor growth (Fig. 2g). The significant decrease in tumor growth by CHK1i treatment allowed for increased overall survival, especially in

**Fig. 2 | Trx1 or TrxR1 depletion and CHK1i show a synergistic interaction in vitro and in vivo. a** Western blots of Trx1 and TrxR1 in H1299 and Calu-6 NSCLC cells after shRNA-mediated depletion. **b** Cellular toxicity assays of the indicated cell lines with the indicated knockdowns to measure cell survival after treatment with CHK1i for 72 h ($n = 4$; biological repeats), error bars represent ±SD. **c, d** Proliferation assays of the indicated cells lines with the indicated knockdowns and after treatment with CHK1i for 24 h ($n = 3$ biological repeats), error bars represent ±SD. **e** Representative photographs of the excised tumors from the indicated groups. **f** The tumor weight from each indicated group at the endpoint of the experiment ($n = 8$; individual tumors in each group; error bars represent ±SD). **g** Tumor growth curves in the indicated groups ($n = 10$ animals for each group), error bars represent ±SD. **h** Survival of the indicated groups as analyzed by Kaplan–Meier analysis. (number of animals in each group is indicated in figure in parentheses). **i** Representative TUNEL images of the tissue excised from the indicated groups treated with or without CHK1i. T1-T3 are the number of tumor tissue. Scale bar, 100 μm. **j** Bar graph showing relative TUNEL intensity in the indicated groups ($n = 3$ individual tumors from each group; error bars represent ±SD). [Statistical information: Data are represented as mean value ± SD. The *p-value*s were calculated using one way ANOVA for multiple comparison; **$p ≤ 0.005$; ***$p ≤ 0.0001$; ****$p < 0.0001$].

animals bearing tumors form Trx1- or TrxR1-depleted cells (Fig. 2h). Of note, a similar body weight among the various groups indicated no apparent systematic toxicity following CHK1i administration during treatment (Supplementary Fig. S4b). We then determined the effect of the different treatments on cancer cell death by conducting H&E staining (Supplementary Fig. S4c) and the TUNEL assay in xenograft tumor tissue. A significantly higher degree of apoptosis as measured by TUNEL assays (Fig. 2I, j) in the group treated with both CHK1i and Trx1 KD or TrxR1 KD compared to the monotherapies. In summary, both in vitro and in vivo findings suggest a potent and significant synergistic interaction between Trx1 or TrxR1 depletion and CHK1 inhibition.

### Trx1/TrxR1 KD in combination with CHK1i leads to profound RS

Cells with high RS depend on ATR/CHK1 for survival, which renders cells sensitive to ATR/CHK1 inhibitors[35–37]. To test the hypothesis that Trx1 or TrxR1 depletion increases RS and activates ATR/CHK1, thereby leading to the increased sensitivity of the CHK1i, we first assessed by Western blot the phosphorylation of ATR/CHK1, RPA32 (p-RPA32) and histone protein H2AX (γH2AX), markers of RS and/or DNA double-strand breaks (DSBs). We found that Trx1 or TrxR1 depletion distinctly increased p-CHK1(S345) and p-ATR (1989) and p-RPA32 and γH2AX compared with shCON, an indication of increased spontaneous RS (Supplementary Fig. S5a). In addition, Trx1 or TrxR1 depletion also led to an increase in chromatin-associated RPA2, p-RPA32 and γH2AX (Supplementary Fig. S5b). Furthermore, nuclear intensity quantification of RPA32, pRPA32 and γ-H2AX together with foci formation revealed significantly increased nuclear intensities and nuclear foci in Trx1- or TrxR1-depleted cells (Supplementary Fig. S5c–h). Supporting these findings, Trx1 or TrxR1 depletion-induced RS resulted in the accumulation of cells in the S phase as detected by flow cytometry (Supplementary Fig. S5i, j, FACS gating strategies are presented in supplementary file, Supplementary Fig. S17). In contrast, Trx1 or TrxR1 inhibition has much less impact in RS untransformed lung cells (Supplementary Fig. S5k). Thus, Trx1 or TrxR1 depletion results in spontaneous RS and DSBs. Of note, Trx1 or TrxR1 KD had no obvious impact on RS (Supplementary Fig. S6a–d) and cytotoxicity (Supplementary Fig. S6e) in the A549 cells that have an intact GSH system (Supplementary Fig. S6).

Contingent with the synergistic inhibition findings (Fig. 2), we sought to determine the effect of the combinatorial inhibition of the Trx system and CHK1 on RS. CHK1i exposure in Trx1 or TrxR1-depleted cells further intensified RS compared to shCON cells (Fig. 3a). Moreover, immunostaining showed further increases in nuclear intensities of p-RPA32 and γH2AX (Fig. 3b, c) and nuclear foci (Fig. 3d–f), following CHK1i treatment, particularly in Trx1 or TrxR1-depleted cells. Neutral comet assays also suggested a greater increase in DSBs in Trx1 or TrxR1-depleted cells following CHK1i treatment compared to shCON cells (Fig. 3g, h). Of note, Trx1 or TrxR1-depleted cells treated with the CHK1i exhibited a significantly higher proportion of cells with pan γH2AX staining, compared to CHK1 inhibition and/or Trx1 or TrxR1 depletion alone (Fig. 3i). In support of the hypothesis that increased RS and DSBs lead to the cell death, we noticed a significant increases in

higher nuclear fragmentation following CHK1i treatment, a typical cell death caused by mitotic catastrophe particularly in Trx1 or TrxR1-depleted cells (Fig. 3j, k). Lastly, cells with depleted Trx1 or TrxR1 showed significantly higher TUNEL staining compared to depletion or CHK1 inhibition alone (Fig. 3l, m). Of note, the extent of RS and DNA damage in Trx1 or TrxR1 KD tumors treated with CHK1i that was described in Fig. 2 were also profound (Supplementary Fig. S7). Collectively, inhibition of CHK1 coupled with Trx1 or TrxR1 depletion induces a greater increase in the level of RS and unrepaired DSBs compared to either inhibition alone, leading to cell death.

### ROS does not play a critical role in Trx1 KD-induced RS or the synergy between Trx1 KD and CHK1 inhibition

The Trx system protects cells by its antioxidant activity via removal of peroxides and ROS through electron donation to downstream thioredoxin peroxidases, such as peroxiredoxins (Prxs). Given that ROS can induce DNA damage, ROS might be responsible for the increased RS in Trx1 or TrxR1-depleted cells. Thus, we measured the levels of intracellular ROS using the fluorescent probe CM-H$^2$DCFDA that is commonly employed and react with several ROS including hydrogen peroxide, hydroxyl radicals and peroxinitrite[38] and found that Trx1 or TrxR1 depletion increased the accumulation of intracellular ROS as indicated by a higher DCFDA signal (Fig. 4a). Simultaneously, we used two different ROS scavenger N-acetyl cysteine (NAC) and α-tocopherol (vitamin E) to scavenge intracellular ROS induced by Trx1 or TrxR1 depletion to the basal level ROS as compared to shCON. NAC is frequently employed as a source of sulfhydryl groups to cells as an acetylated precursor of reduced GSH. NAC can also interact directly with ROS and nitrogen species because it is a scavenger of oxygen free radicals[39]. α-tocopherol can significantly scavenge both peroxyl and singlet oxygen species[40,41], both of which can cause DNA damage[42,43]. Therefore, we used NAC and α-tocopherol as scavengers to remove the DNA damage-inducing ROS that are produced in cells upon inhibition of the Trx system. By titration, we found that -1 mM NAC and -10 μM of α-tocopherol was sufficient to bring the Trx1 or TrxR1 depletion induced-ROS level to almost basal levels (Fig. 4a and Supplementary Fig S8a). We, then measured the changes in lipid peroxidation as secondary verification for ROS scavenging in cells treated with NAC or α-tocopherol following Trx1 or TrxR1 KD. Knockdown-induced lipid peroxidation was brought down significantly close to shCON groups (Fig. 4b and Supplementary Fig. S8b). Surprisingly, treatment with NAC or α-tocopherol failed to mitigate Trx1 or TrxR1 depletion-induced RS, as reflected by no profound changes in the RS markers (Fig. 4c, d). Additionally, no significant change was found in Trx1 or TrxR1 depletion-induced p-RPA32 foci in the cells with or without NAC (Fig. 4e). Interestingly, we observed a decrease in γH2AX foci induced by Trx1 or TrxR1 depletion following NAC treatment, relative to untreated cells (Fig. 4f). This might be caused by the fact that apoptosis can also lead to increased γH2AX, and ROS scavenger treatment might have an impact on apoptosis. Consistent with a lack of impact on Trx1 or TrxR1 depletion-induced RS, NAC intervention had no obvious impact on Trx1 or TrxR1 depletion-induced cell growth repression (Fig. 4g, h) and failed to decrease CHK1i sensitivity following Trx1 or

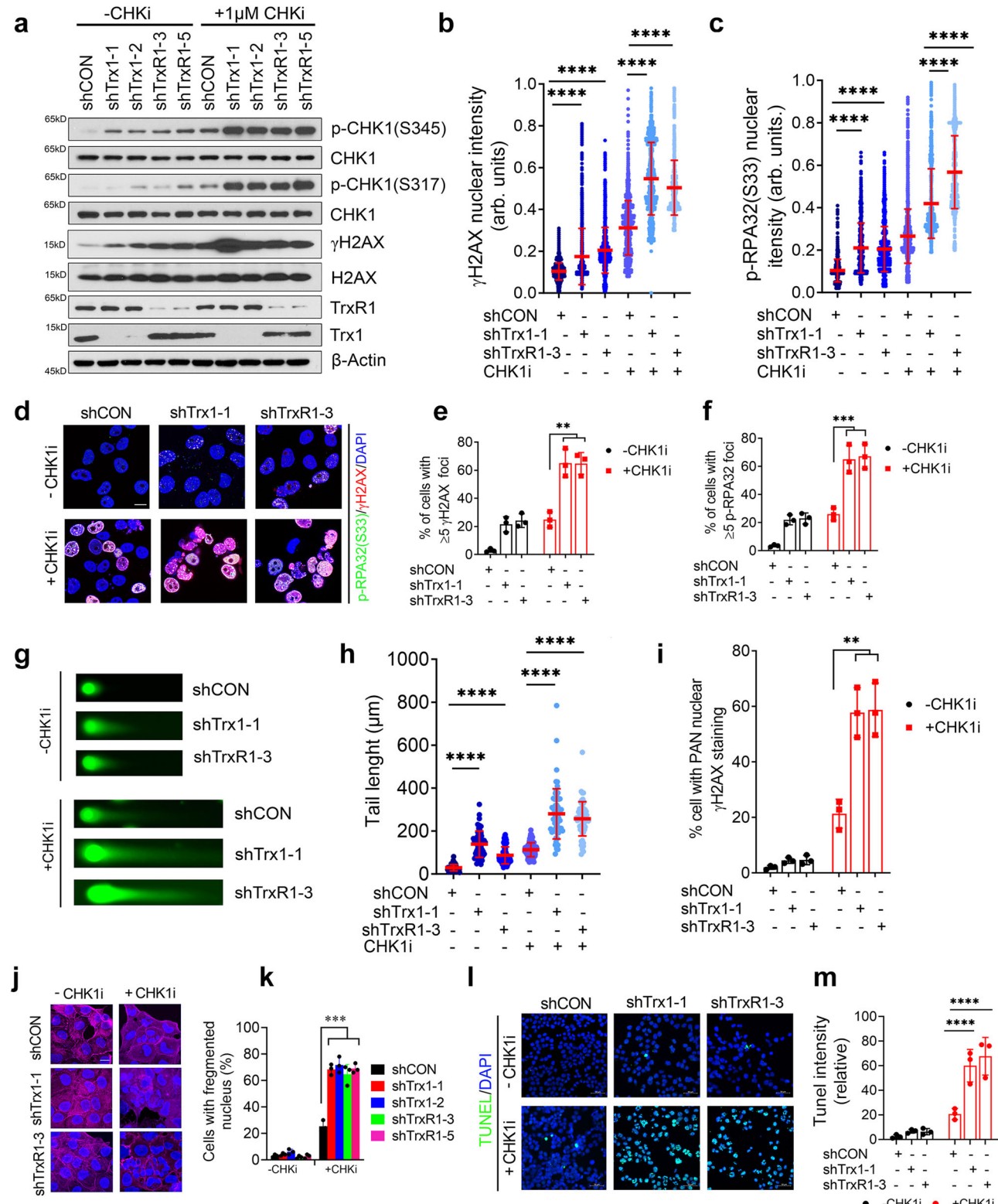

TrxR1 depletion (Fig. 4i, j). Collectively, our results suggest that Trx1 or TrxR1 depletion-induced ROS is not responsible for RS and ROS scavenging does not mitigate the enhanced sensitivity to CHK1i following Trx1 or TrxR1 KD.

## Trx1/TrxR1 KD impairs replication fork elongation due to a depleted dNTP pool

To determine the mechanisms by which Trx1 or TrxR1 depletion increases RS, we first determined the rate of DNA synthesis during S phase progression via a BrdU incorporation assay. By doing so, we found an increased accumulation of cells in early S phase, which typically occurs when cells have a deficient dNTP pool (Supplementary Fig. S9a, b, FACS gating strategies are presented in Supplementary Fig. S17). Indeed, we observed a significant decrease in the dNTP pool following Trx1 or TrxR1 depletion (Fig. 5a–d). To characterize the fate of replication forks caused by a disbalanced dNTP pool, we analyzed the replication elongation, initiation and termination that are frequently affected during RS using the DNA fiber technique[44]. We found

**Fig. 3 | Trx1 or TrxR1 depletion in combination with CHK1i leads to profound RS, replication fork collapse and cell death. a** Western blot analysis of markers of RS and DNA damage in H1299 cells treated with 1 μM CHK1i for 1 h. **b, c** The nuclear intensities of individual cells of γH2AX and p-RPA32(S33) following CHK1i treatment ($n \geq 1500$ cells from 3 biological repeats), error bars represent ±SD. **d** Representative immunofluorescence images of γH2AX- and p-RPA32(S33)-positive foci in H1299 cells. Scale bar, 50 μm. The percentage of H1299 cells with ≥5 foci of γH2AX (**e**) and p-RPA32(S33) (**f**) following CHK1i treatment for 1 h. ($n \geq 100$ cells/repeat from 3 biological repeats; error bars represent ±SD). Representative images from neutral comet assays to measure the degree of DNA double strand breaks in H1299 cells treated with 1 μM CHK1i for 2 h (**g**) and the quantitation of their tail lengths (**h**) ($n \geq 50$ cells/repeat from 3 biological repeats), error bars represent ±SD.

**i** Percentage of cells with Pan-γH2AX staining after CHK1i treatment ($n \geq 100$ cells/repeat from 3 biological repeats; error bars represent ±SD). Representative immunofluorescent images of nuclear fragmentation (DAPI staining) and F-actin staining (Phalloidin) (**j**) after CHK1i treatment for 24 h and the quantitation of the percentage of cells with nuclear fragmentation (**k**) ($n = \geq 100$ cells/repeat from 3 biological repeats; error bars represent ±SD) Scale bar, 50 μm. TUNEL staining (**l**) and its quantitation (**m**) of H1299 cells treated with CHK1i (1 μM) for 6 h. ($n \geq 100$ cells/repeat from 3 biological repeats; error bars represent ±SD) Scale bar, 50 μm. [Statistical information: Data are represented as mean value ± SD. The *p-value*s were calculated using one way ANOVA for multiple comparison. Red line in dot plot (**b, c** and **h**) indicates mean; **$p \leq 0.005$; ***$p \leq 0.0001$; ****$p < 0.0001$; ns non-significant].

that Trx1 or TrxR1 depletion significantly impeded replication fork progression (Fig. 5e, f), which resulted in a higher rate of fork termination and simultaneous initiation of new origin firings, especially in Trx1 or TrxR1-depleted conditions (Fig. 5g). We validated the contribution of a reduced dNTP pool to replication fork stalling following Trx1 or TrxR1 depletion via reconstitution with extracellular dNTPs. This repletion significantly restored replication fork elongation in Trx1 or TrxR1- depleted cells (Fig. 5h). Similarly, exogenous addition of dNTPs remarkably mitigated Trx1 or TrxR1 depletion-induced RS, as indicated by decreased p-CHK1, p-RPA2 and γ-H2AX levels (Fig. 5i). Of note, considering the ROS-independent RS induction in Trx1 or TrxR1-depleted cells, the results of DNA fiber assays also suggested no mitigation of replication elongation or decreases in new origin firing upon NAC treatment in Trx1 or TrxR1-depleted cells. (Fig. 5–l). Taken together, we conclude that Trx1 and TrxR1 depletion-induced RS is tightly linked with an imbalanced dNTP pool.

## RRM1 cysteine oxidation is required for Trx1 KD-induced RS

RNR, the rate limiting enzyme for dNTP synthesis, relies on both internal and external electron transfer flow to maintain redox cycling; *i.e.*, enzymatic activity[16]. Reportedly, RRM1 is a substrate of the Trx system in *E. coli* and *Saccharomyces cerevisiae*[15,45,46] although it has not yet been shown in mammalian cells. Given the impaired DNA replication fork progress and imbalanced dNTP pool in Trx1 or TrxR1-depleted cells (Fig. 5a–d), we hypothesized that Trx system inhibition-induced RS is associated with RRM1 enzymatic inactivity in human cells. To test our hypothesis, we first determined if Trx1 depletion alters the levels of oxidized RRM1 using iodoacetamide (IAA) to alkylate free thiols. IAA modification in protein thiols (reduced form) results in thiol alkylation and adds one negative charge per thiol modified. Consequently, reduced Trx1 with negative charges can be separated from the oxidized form due to higher negative charge during non-reducing SDS gel electrophoresis[47]. Scrambled shRNA (shCON)-treated cells predominantly exhibited a faster migrating form of RRM1, whereas depletion of Trx1 resulted in accumulation of slower migrating RRM1 (Fig. 6a). Notably, protein extract reduction with DTT prior to gel loading led to the complete disappearance of the upper band, indicating that the slower migrating band represents the oxidized form of RRM1 while the lower band is the reduced form (Fig. 6a). This indicates that Trx system inhibition interrupted the electron flow between Trx1 and its substrate RRM1. Thus, Trx1 depletion-induced an alteration in electron flow, leading to the accumulation of oxidized RRM1. Of note, NAC treatment failed to exert any notable impact on RRM1 oxidation in the Trx1 depletion group, again indicating a ROS-independent change in RRM1 redox recycling caused by the depletion (Fig. 6b).

To determine if RRM1 oxidation contributes to Trx system inhibition-induced RS, we next determined the linkage between RRM1 oxidation with RS using wild-type or redox-inactive Trx1 mutants. The functional activity of Trx1 is critically associated with the conserved disulfide motif (Cys$^{32}$-Gly-Pro-Cys$^{35}$ of hTrx1) in the active site, which is

indispensable for Trx1 redox activity. We exogenously expressed wild-type Trx1 (Flag-Trx1-WT) and redox mutant Trx1 (Flag-Trx1(C32S) and Flag-Trx1(C35S)) in cells expressing 3′UTR shRNA of Trx1. Trx1-WT re-expression in Trx1-depleted cells rescued the effect on oxidized RRM1 accumulation in response to Trx1 depletion, whereas redox inactive Trx1 expression failed to rescue this effect (Fig. 6c). Correspondingly, Trx1-WT abolished the increase in phosphorylation of RS markers triggered by Trx1 depletion, whereas redox mutant Trx1 (Trx1 C32S, C35S) showed no significant impact on this readout (Fig. 6d). Additionally, unlike the redox mutants, Trx1-WT expression was associated with a significant repletion of the dNTP pool (Fig. 6e), and it facilitated S-phase progression (Fig. 6f). FACS gating strategies are presented in supplementary Fig. S17. Scrutiny of DNA fibers revealed that WT-Trx1 significantly restored the Trx1 depletion-mediated replication impediment; however, redox mutants Trx1 (Trx1 C32S, C35S) failed to mitigate the slower replication rate (Fig. 6g). Reinstatement of DNA replication and dNTP pool replenishment significantly diminished RS-associated DNA damage (Fig. 6d, h). Finally, we determined the CHK1i sensitivity in Trx1-depleted cells following WT or redox mutant Trx1 (C32S, C35S) expression. We noted a significant recovery in cell proliferation in cells re-expressing WT Trx1, whereas expression of the redox mutants failed to impact cell proliferation (Supplementary Fig. S10a). Similarly, Trx1-WT-expression decreased CHK1i sensitivity, whereas expression of the redox mutants failed to curtail CHK1i sensitivity in Trx1-depleted cells (Supplementary Fig. S10b). Collectively, results suggests that RRM1 oxidation is responsible for Trx1 depletion-mediated RS that leads to enhanced sensitivity to CHK1i.

Reportedly, in the R1 (equivalent to human RRM1) subunit of *E. coli* RNR, a narrow entrance to the catalytic site sterically prohibits a direct reduction of the active-site disulfide by Trx or Grx. Five residues in R1 are essential for its activity at the active sites; namely, Cys225, Asn437, Cys439, Glu441 and Cys462 in *E. coli*[18]. R2 subunit-initiated transfer of tyrosyl radical [Tyr122 (Y•)] to Cys462 in R1 facilitates the formation of first disulfide bond formation within the R1 active site between Cys225 and Cys462 and is subsequently transposed to a C-terminal disulfide bond between C754 and C759 via a long-range electron transfer pathway. Finally, disulfide bond formation between two C-terminal cysteines (Cys754 and Cys759 in *E. coli*, Cys787 and Cys790 in mammals) then acts as substrate for the external Trx system to be reduced to facilitate the conversion (Fig. 6i). It is well established that most eukaryotes and prokaryotes exhibit two highly conserved cysteine residues in the C-terminal regions at Cys787 and Cys790 of RRM1, which are required for the reduction of the catalytic site. Thus, we use RRM1 redox mutants (Myc-RRM1 C787S and C790S) that are devoid of both internal and external electron exchange capabilities due to serine substitution to investigate its impact on Trx1 depletion-induced RS. Additionally, we also mutated another cysteine (Cys779) proximal to the C-terminal tail that is closer to the Trx1 substrate Cys787-X-X-Cys790. The site Cys779 might be involved in regulation of RNR activity by other mechanism, but it has no direct role in exchange of electrons through the external Trx system.

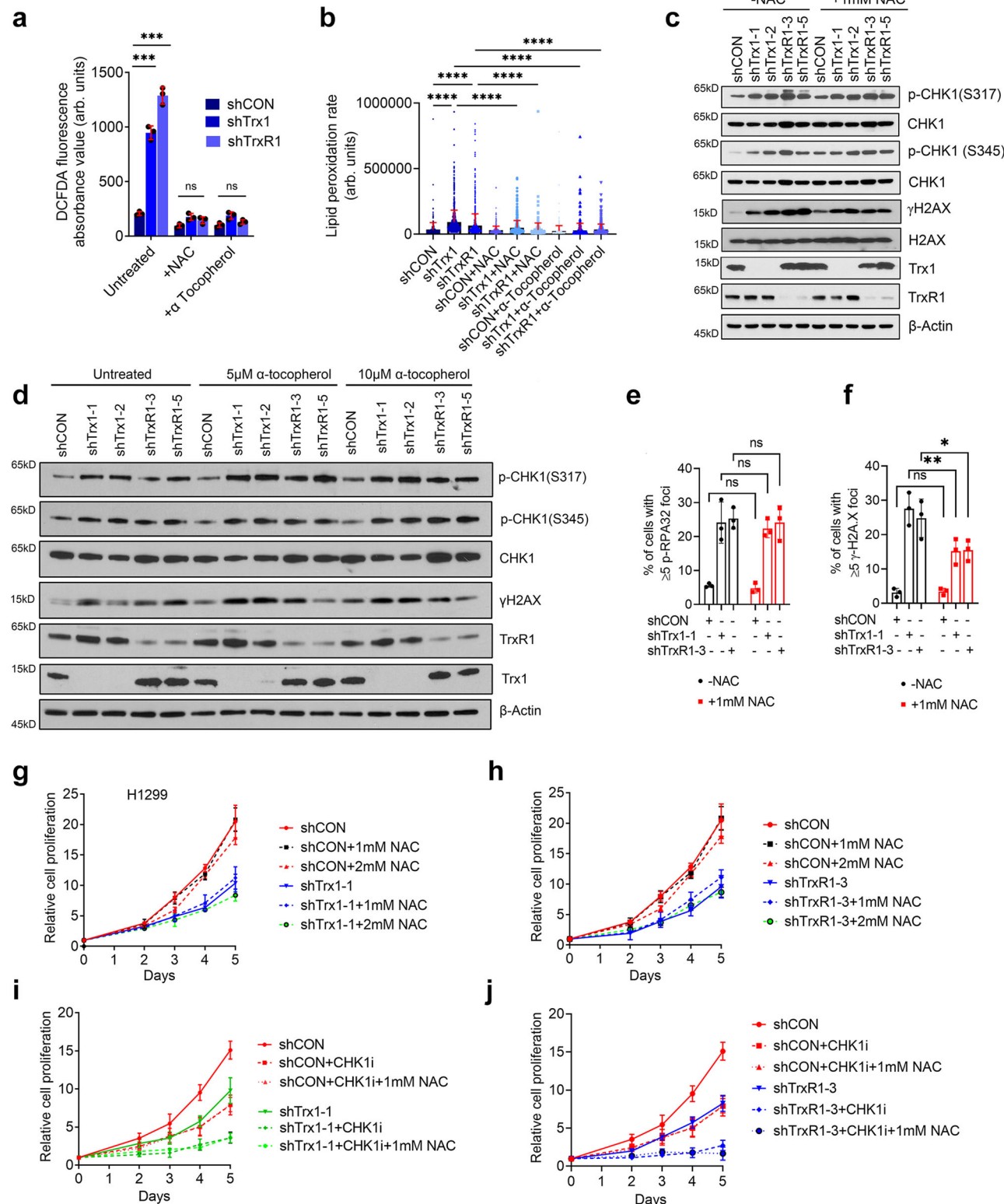

**Fig. 4 | ROS is not responsible for Trx1 or TrxR1 depletion-induced RS.**
**a** Intracellular ROS levels in H1299 cells treated with 1 mM NAC and α-tocopherol (10 μM) as measured by DCFDA staining ($n = 3$ biological repeats; error bars represent ±SD). **b** Bar graph showing changes in lipid peroxidation in indicated groups treated with or without NAC and α-tocopherol ($n ≥ 1000$ cells/repeat from 2 biological repeats; error bars represent ±SD). **c, d** Representative western blots of the indicated RS markers in the experimental condition after NAC and α-tocopherol treatment of H1299 cells. Percent of H1299 cells with ≥5 foci of p-RAP32(S33) (**e**) and

γH2AX (**f**) after NAC treatment (1 mM). ($n ≥ 100$ cells/repeat from 3 biological repeats; error bars represent ±SD). Relative cell proliferation of H1299 cells depleted for Trx1 (**g**) or TrxR1 (**h**) after NAC treatment (1 mM) ($n = 2$ biological repeats; error bars represent ±SD). Relative cell proliferation of H1299 cells after Trx1 depletion (**i**) or TrxR1 depletion (**j**) and CHK1i treatment and NAC treatment (1 mM) ($n = 2$ biological repeats; error bars represent ±SD). [Statistical information: Data are represented as mean value ± SD. The *p-value*s were calculated using one way ANOVA for multiple comparison; *$p ≤ 0.05$; **$p ≤ 0.001$; ns non-significant*].

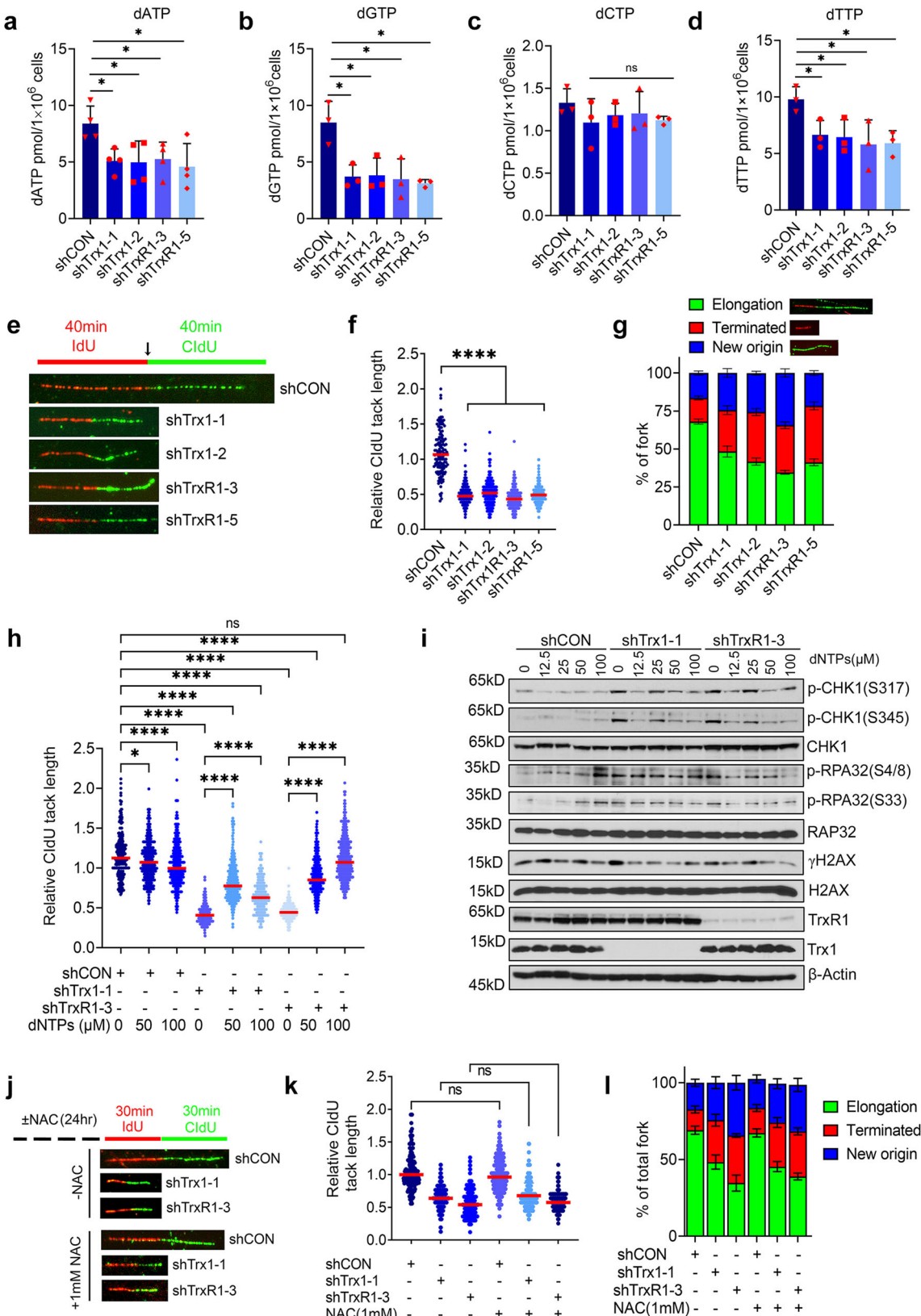

We started with assessment of RRM1 reduction-oxidation following Cys→Ser substitution in the previously indicated cystine residues. We found that single mutations at Cys787S and Cys790S and double mutations (Cys787S/Cys790S) increased the accumulation of RRM1 oxidation (Fig. 6j). However, RRM1-WT-and a single mutation at C779S did not show detectable amount of oxidized form accumulation.

Interestingly, a single Cys substitution at C779S does not result in accumulation of oxidized RRM1; however, the C779S mutation in combination with Trx1 substrate sites Cys787 and/or Cys790 (C779S/C787S; C779S/C790S and C779S/C787S/C790S) leads to accumulation of oxidized RRM1. This indicates the redundancy of Cys779 in redox recycling of RRM1 and no participation in electron exchange via

**Fig. 5 | Trx1 or TrxR1 depletion abrogates dNTP production and interrupts replication fork elongation.** Levels of dATP (**a**), dGTP (**b**), dCTP (**c**) and dTTP (**d**) after Trx1 or TrxR1 depletion in H1299 cells ($n = 3$ biological repeats; error bars represent ±SD). **e** Representative image of DNA fiber tracks in the indicated groups after 40 min pulses with IdU and CldU. **f** Replication fork progression as measured by relative CldU track lengths (CldU to IdU ratio) of the groups in **e**. A CldU-to-IdU ratio closer to -1 one indicates a normal speed of replication fork with a ratio <1 indicating a slower progression of the fork. 100–200 fibers ($n = 3$ biological repeats; ≥100–200 fibers/repeat) were counted per conditions/experiment and the ratio of each fiber is represented in dot plot. The median line is indicated in red. **g** The percentage of replication fork progression, elongation termination and new origin firing in the indicated groups. ($n = 3$ biological repeats; ≥100 fibers/repeat; error bars represent ±SD). **h** Effects on replication fork progression after exogenous dNTP treatment in the indicated groups ($n = 3$ biological repeats; ≥100–200 fibers/repeat). **i** Representative western blots of the indicated RS markers after addition of exogenous dNTPs to H1299 cells that have been depleted for Trx1 or TrxR1. **j** Representative images of DNA fibers from the indicated cells treated with 1 mM NAC for 24 h. **k** The effect of NAC treatment on replication fork progression in the indicated cells. ($n = 3$ biological repeats; ≥100–200 fibers/repeat). **l** The effect of NAC treatment on replication fork progression, elongation termination and new origin firing in the indicated groups ($n = 3$ biological repeats; ≥100 fibers/repeat; error bars represent ±SD). [Statistical information: Data are presented as mean value ± SD. The *p-value*s were calculated using one way ANOVA for multiple comparison. Red line in dot plot (**f**, **h** and **k**) indicates mean; *≤0.05; **** $p < 0.0001$; ns non-significant].

external Trx system (Fig. 6j). Reportedly, overexpression of wild-type RRM1 in Trx1-deficient *S. cerevisiae* rescued Trx deficiency-linked phenotypes and decreased the oxidized RRM1 due to a simultaneous increase in RRM1 reduction status[23]. Similarly in human cells, we observed that exogenous Myc RRM1-WT and C779S does not accumulate in their oxidized form under the experimental condition, suggesting their efficient redox recycling (Fig. 6k). Consistent with these results, we found expression of exogenous RRM1-WT reduced Trx1-depletion-induced RS whereas a RRM1-redox mutation failed to do so, which is consistent with the previous observation in yeast showing that RRM1-WT, but not a redox mutant, rescues the RS in Trx1-depleted cells (Fig. 6l). The C779S expression was found to have no impact on the oxidation level of RRM1 and the Trx1 depletion-induced RS, like the effect of RRM1 WT. In summary, our study suggests that Trx1 KD leads to RS and slows replication forks velocity by increasing RRM1 oxidation and thus decreasing the dNTP pool in mammalian cells. Therefore, our finding establishes that RRM1 serves as one of the direct substrates of Trx1, which undergoes redox cycling via the Trx system, and C787 and/or C790 oxidation of RRM1 are critical for Trx1 KD-induced RS.

## CHK1 inhibition disrupts Trx1/TrxR1 depletion-mediated E2F1-RRM2 pathway

As previously reported, the CHK1-E2F1-RRM2 pathway is activated under conditions of RS[28]. We found that Trx system interruption triggered ATR/CHK1 activation (Supplementary Fig. S5a–j) and increased the expression of RRM2, which is accompanied by overexpression of the transcription factor E2F1 (Fig. 7a). CHK1 inhibition abrogated Trx1 or TrxR1 depletion-induced E2F1-RRM2 induction (Fig. 7b). Mechanistically, functional RNR depends on both RRM1 and RRM2, and RNR activity require long-range electron transfer from RRM2 to RRM1 to produce dNTPs (Figs. 6i and 7c). In support of the result that CHK1i limits RRM2 expression in Trx1 or TrxR1-depleted cells, CHKi leads to a further decline in dNTP production in cells depleted of Trx1 or TrxR1 alone (Fig. 7d and Supplementary Fig. S11). Furthermore, CHK1i significantly decreased DNA replication fork velocity in Trx1 or TrxR1-depleted cells (Fig. 7e, f). Therefore, RRM2 inhibition by CHKi further impaired RNR activity and dNTP biosynthesis, leading to a slow replication fork progression in cells with Trx system disruption.

To validate our hypothesis that CHK1i-induced inhibition of RRM2 expression indeed impairs RNR activity, we determined the impact of CHK1i on RRM1 oxidation in the cells with or without Trx1 depletion. The cells were treated with the CHK1i or hydroxyurea (HU), an established RNR inhibitor that inhibits RRM2 via scavenging the iron-tyrosyl free radical[48]. HU significantly decreased the accumulation of oxidized RRM1 triggered by Trx1 depletion (Fig. 7g, h, left panel). Similarly, CHK1i also abrogated Trx1 depletion-induced RRM1 oxidation (Fig. 7g, h, right panel). Mechanistically, a long-range intermolecular electron transfer into RRM1 (R1 subunit) is initiated at Tyr122 cofactor (Y•) at a di-nuclear iron cluster ($Fe^{III}Fe^{III}$) in RRM2 (R2 subunit) (Fig. 6i). Thus, a RRM2-localized radical initiates the formation of the first disulfide bond in RRM1. When RRM2 is inhibited by CHK1i or HU, the electron transfer flow was interrupted at the initial step; thus, leading to the decrease in the oxidized RRM1.

To further investigate the CHK1i-mediated RRM2 inhibition and its effect on the long-range intermolecular electron transfer to RRM1, we next determined the RRM1 oxidation status in CHK1i- or HU-treated cells expressing Myc-RRM1 WT and its redox mutants. As with HU, CHK1i significantly abrogated the accumulation of the oxidized form of the redox mutants of RRM1, which supports the critical role of RRM2 in the initiation of electron transfer within RRM1 (Fig. 7i, j). Therefore, simultaneous inhibition of RRM2 by CHKi in the absence of the external electron donor Trx1 renders RNR inactive to a greater extent than CHK1i treatment alone, leading to a drastic loss of the dNTP pool that further sensitizes the anti-tumor effects of the drug.

## The TrxR1 inhibitor AUR exerts an antitumor effect via synergistic interaction with CHK1i in NSCLC cells

Inhibitors of the Trx system have been developed, but most of them have failed to translate to the clinic due to high toxicity and insufficient targeting efficacy. However, AUR, a gold-containing triethyl phosphine compound that was originally approved to treat rheumatoid arthritis, is a well-known inhibitor of TrxR. Therefore, we next determined if AUR has a similar impact on RS and CHK1i sensitivity as Trx1 or TrxR1 depletion. We noted a significant dose-dependent attenuation in cell growth in several NSCLC cells following AUR treatment (Supplementary Fig. S12a). Additionally, AUR exacerbated CHK1-mediated cell killing in a panel of different NSCLC cell lines and severely limited cell viability (Fig. 8a). Concordantly, combined AUR and CHK1i treatment significantly decreases cell proliferation compared to either treatment alone (Supplementary Fig. S12b). An ATRi also shows a similar synergistic interaction with AUR (Supplementary Fig. S12c). To investigate whether the combination of AUR and CHK1i is synergistic or additive in nature, we used SynergyFinder 2.0 to generate an inhibitory dose response matrix of AUR and CHK1i and found a dose-dependent increase in cell toxicity for the combinatorial treatment (Supplementary Fig. S12d). The drug interactive algorithm utilizes four different synergy models (HAS, Bliss, Loewe and ZIP) as a reference and uses their extensions to calculate the synergy score for combinatorial data of multiple drugs. A synergy scores lower than −10 indicates an antagonistic response and a score between −10 to +10 indicates an additive effect. A synergy score >+10 indicates a synergistic interaction between the given drugs. The synergy score anticipated by all four different synergy models, ZIP (score: 16.54) (Fig. 8b), Loewe (score: 20.94) (Fig. 8c), Bliss (score: 16.42) (Fig. 8d) and HSA (score: 25.9) (Fig. 8e), generated a score greater than +10; indicating a synergistic interaction between AUR and CHK1i. AUR exerted a massive synergistic potency shift of 34752.01-fold against CHK1i and CHK1i displayed synergistic potency shift of 7.025-fold against AUR. However, AUR and CHK1i displayed only a 0.15-fold negative cooperativity in combination (Supplementary Fig. S12e).

Furthermore, we validated our in vitro combinatorial treatment strategies using an in vivo xenograft model. We treated tumor-bearing

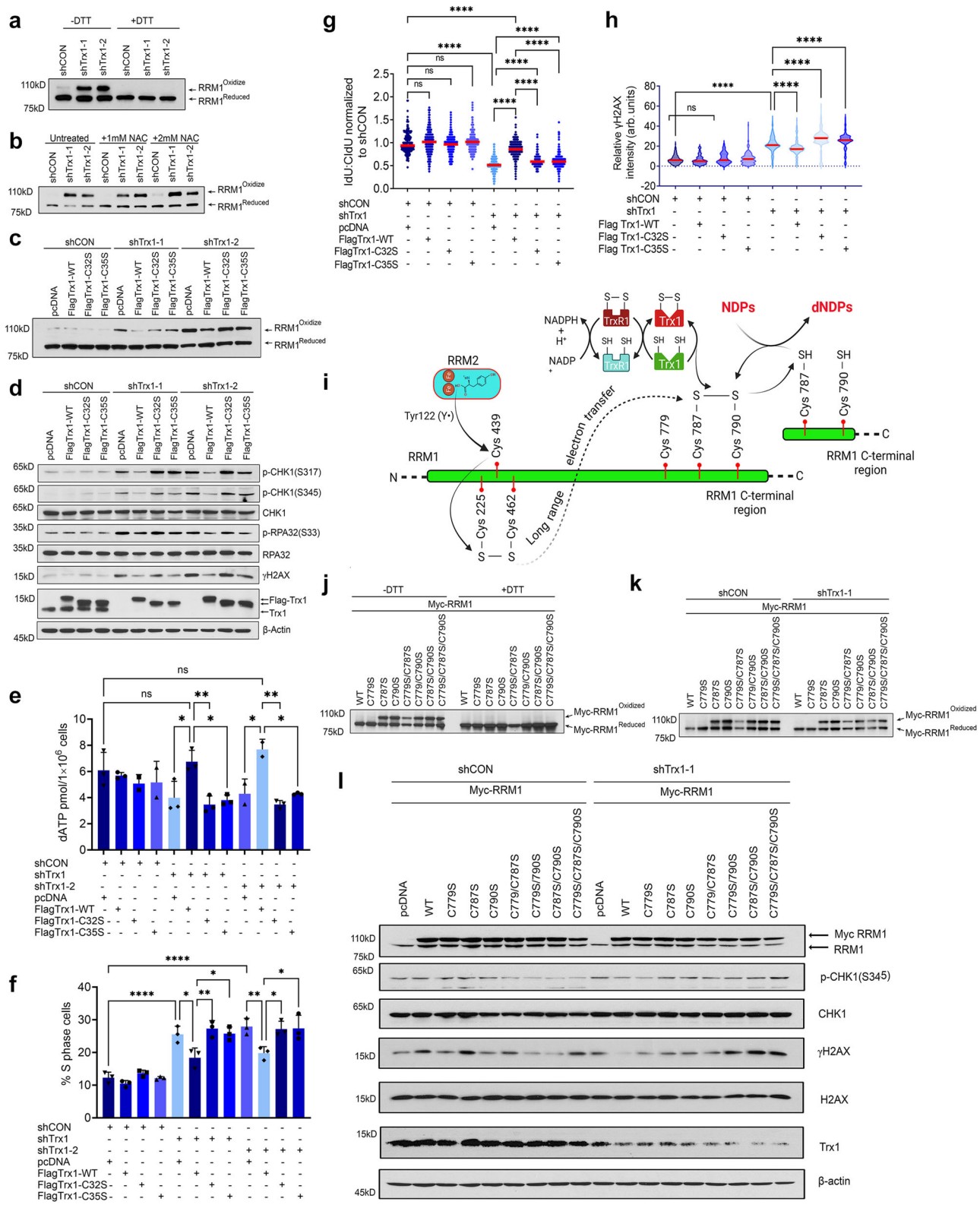

animals with AUR (10 mg/kg) and CHK1i (25 mg/kg) (Supplementary Fig. S13a). Animals treated with the combined AUR and CHK1i showed significant reduction in tumor growth (Fig. 8f), leading to increased overall survival (Fig. 8g). Overall tumor burden which was indicated by tumor size and weight was significantly reduced in combinatorial treatment group (Fig. 8h, i). Of note, no significant loss in overall body weight indicates that the mice tolerated the combination therapy with no apparent toxicity (Supplementary Fig. S13b). Additionally, we also

assessed the changes in RS, DNA damage and apoptosis across the treatment groups and found that tumors treated with the combined inhibitors had greater RS, DNA damage and apoptosis (Supplementary Fig. S13c–f).

Because of the fidelity of patient-derived xenograft (PDX) models to clinical scenarios, these models have been widely employed in exploring therapeutic effects. To further validate our results from the xenograft model, we utilized the PDX (PDX-72) of NSCLC[49] to

**Fig. 6 | RRM1 cysteine oxidation is required for Trx1 KD-induced RS. a** Representative western blots of oxidized and reduced RRM1 from H1299 cells under nondenaturing (−DTT) or denaturing (+DTT) conditions following iodoacetamide carboxyamidomethylation and dephosphorylation of the extract. **b** Representative western blots of oxidized and reduced RRM1 from the indicated H1299 cells and after treatment with 1 or 2 mM NAC for 24 h. **c** Representative western blots of oxidized and reduced RRM1 from Trx1-depleted H1299 cells transfected with Trx1-WT or Trx1 redox mutants. **d** Representative western blots of RS markers in Trx1-depleted H1299 cells transfected with Trx1-WT or Trx1 redox mutants. **e** The levels of dATP in the indicated group of cells (*n* = 3 biological repeats; error bars represent ±SD). **f** The percentage of cells in S phase among the indicated group of cells (*n* = 3 biological repeats; error bars represent ±SD). **g** The effect on replication fork progression in the indicated group of H1299 cells (*n* = 3 biological repeats; ≥100 fibers/repeat). **h** Violin plots of relative γH2AX intensity among the indicated groups of cells (*n* = 2 biological repeats; ≥500 cells/repeat). **i** A schematic diagram *(Created with BioRender.com released under a Creative Commons Attribution-NonCommercial-NoDerivs 4.0 International license)* illustrating the electron transfer flow between RRM2-RRM1 and RRM1 and the Trx system. **j** Representative western blots of oxidized and reduced mutant RRM1 expression in non-reducing (−DTT) and reducing (+DTT) conditions. **k** Representative western blots of oxidized and reduced mutant RRM1 in the indicated cells with or without Trx1 depletion. **l** Representative western blots of the indicated proteins in Trx1-depleted cells expressing various mutant forms of RRM1. [Statistical information: Data are presented as mean value ± SD. The *p-value*s were calculated using one way ANOVA for multiple comparison. Red line in dot plot (**g**) and violine plots (**h**) indicates mean; *≤0.05; **p ≤ 0.005; ****p < 0.0001; ns non-significant].

determine the synergy of the combinatorial treatment approach. A combination of AUR and CHKi significantly slowed the tumor growth rate compared to monotherapy (Fig. 8j), resulting in an increased survival rate (Fig. 8k) and an overall smaller tumor burden as assessed by significantly less tumor size and weight (Fig. 8l, m). The treatment outcome of PDX-72 suggests a high sensitivity to the combination. Of note, the sensitivity of PDX-72 to AUR alone suggests a deficiency of Grx-GSH system. Additionally, we also established that the inhibition of PDX-72 growth by the combinatorial treatment is due to greater apoptosis as measured by a TUNEL assay of tumor tissues (Supplementary Fig. S14a–c). Further, the significant impact of the AUR and CHKi combination was further validated by generating PDX model-derived organoids (PDOs). We dissociated the PDX tumor and isolated the human cancer cells via a human cancer cell isolation kit and validated the human cancer cells by human-specific EpCAM (CD326) antibody-based flow cytometry (Supplementary Fig. S14d, FACS gating strategies are presented in Supplementary File, Supplementary Fig. S17). Combinatorial treatment led to a significant increase in cell death as measured by PI uptake in dead cells compared to the monotherapy (Fig. 8n, o). Furthermore, we detected a higher level of cleaved PARP1 in PDOs treated with the combination therapy as compared to untreated PDOs (Supplementary Fig. S14e). To exclude the possibility of off-target effects of AUR, we further validated the results by utilizing the Trx-specific inhibitor PX-12 and found that PX-12 also significantly decreased the organoid formation rate, resulting in smaller organoid formation as compared to untreated or mono treatment (Fig. 8p, q).

Taken together, our results suggest a strong synthetic lethality occurs between CHK1 inhibition and Trx system inhibition in NSCLC cells in vitro and in tumors in vivo.

### AUR triggers RRM1 oxidation and dysregulates the dNTP pool and has a synergy with CHK1i

Finally, we investigated whether the pharmacological inhibition of the Trx system mimics the Trx system depletion-mediated RS-associated phenotypes. We first determined the impact of AUR on RRM1 redox recycling and found accumulation of oxidized RRM1 accompanied with increased oxidized aggregation of its electron doners Trx1 and TrxR1 following AUR treatment (Fig. 9a). Inactivation of RRM1 redox cycling by AUR due to disruption of electron flow abrogates the RNR enzymatic activity, resulting in defective dNTP biogenesis (Fig. 9b). Additionally, AUR treatment increased the production of intracellular ROS (Supplementary Fig. S15a) and resulted in S-phase accumulation of the cells, but NAC co-treatment failed to rescue AUR-induced cell cycle arrest, suggesting a ROS-independent cell cycle arrest (Fig. 9c, d; FACS gating strategies are presented in Supplementary Fig. S17). Additionally, exogenous reconstitution of the dNTPs significantly mitigated AUR-induced impediment of the replication fork (Fig. 9e, f). Consistent with the Trx1 or TrxR1 depletion data, AUR treatment alone increases RS in NSCLC cells (Supplementary Fig. S15b). CHKi1 further intensified the AUR-induced RS and produced massive DNA damage, leading to the accumulation of unpaired DSBs (Fig. 9g–j). By DNA fiber assays we found that combinatorial treatment with CHK1i and AUR exerted a more profound defect in the progression of the replication tracks and resulted in increased replication termination (Fig. 9k, l). Additionally, we assessed the apoptotic efficacy of AUR and CHKi in NSCLC cells via annexin staining and found a significantly higher percentage of apoptotic cells among the combination treatment group compared to single treatment (Supplementary Figs. S15c, d, FACS gating strategies are presented in Supplementary Fig. S17).

We then tested PX-12 and found that it effectively inhibited the proliferation of NSCLC cell lines in a synthetic lethal manner when combined with the CHK1i (Supplementary Fig. S16a). The combined treatment of PX-12 and CHK1i resulted in an increased level of cleaved PARP1 like AUR combination with CHKi (Supplementary Fig. S16b). Additionally, PX-12 treatment led to an increase in the oxidation of RRM1 (Supplementary Fig. S16c). Thus, our data, obtained with PX-12 demonstrates a synthetic interaction with CHK1i in the treatment of NSCLC, which is like what we found in the cells treated with AUR and CHKi. Therefore, there is a synergistic interaction between inhibition of different components of the Trx system and CHK1 in the treatment of NSCLC.

## Discussion

A steady and sufficient supply of dNTPs is required for error-free replication of DNA in proliferating cells. The maintenance of a balanced pool of dNTPs is tightly regulated by RNR and interruption of its enzymatic activity results in insufficient dNTPs and its associated phenotypes, such as RS and S-phase cell accumulation[50,51]. Our study reveals three main findings (Fig. 9m). Firstly, we found that Trx system inhibition-triggered RS relies on Trx system-mediated redox recycling of RRM1 activity in mammalian cells that is critically dependent on the C-terminal cystines of RRM1, but that is independent of ROS. Secondly, inhibition of Trx system-induced elevation in RRM2 by CHK1i terminates its tyrosyl radical production activity and shuts off long-range electron transfer to RRM1. And thirdly, combination of Trx system inhibition and CHK1i can serve as a potent combinational therapy to strikingly reduce tumor growth and prolong survival. And notably, these findings suggest that AUR could be repurposed to treat NSCLC when combined with a CHKi.

The side chain of cysteine (Cys) undergoes both reversible and irreversible redox post-translational modifications (redox-PTMs) and is tightly linked with structural and functional integrity of the proteins. Here we demonstrate that the Trx system acts as an electron donor for RRM1 in mammalian cells and interruption of this system increases RS due to dNTP pool depletion due to inhibition of redox cycling of RRM1. Further, we demonstrate that the depletion or inhibition of the Trx system increases RRM1 oxidation, resulting in slow replication fork progress due to an imbalanced dNTP pool. The exogenous addition of dNTPs rescues the RS induced by interruption of the Trx system, which supports the role of RRM1 redox recycling in DNA precursor synthesis. Consistent with previous reports showing that replication initiation is important for the rescue of collapsed replication forks that are

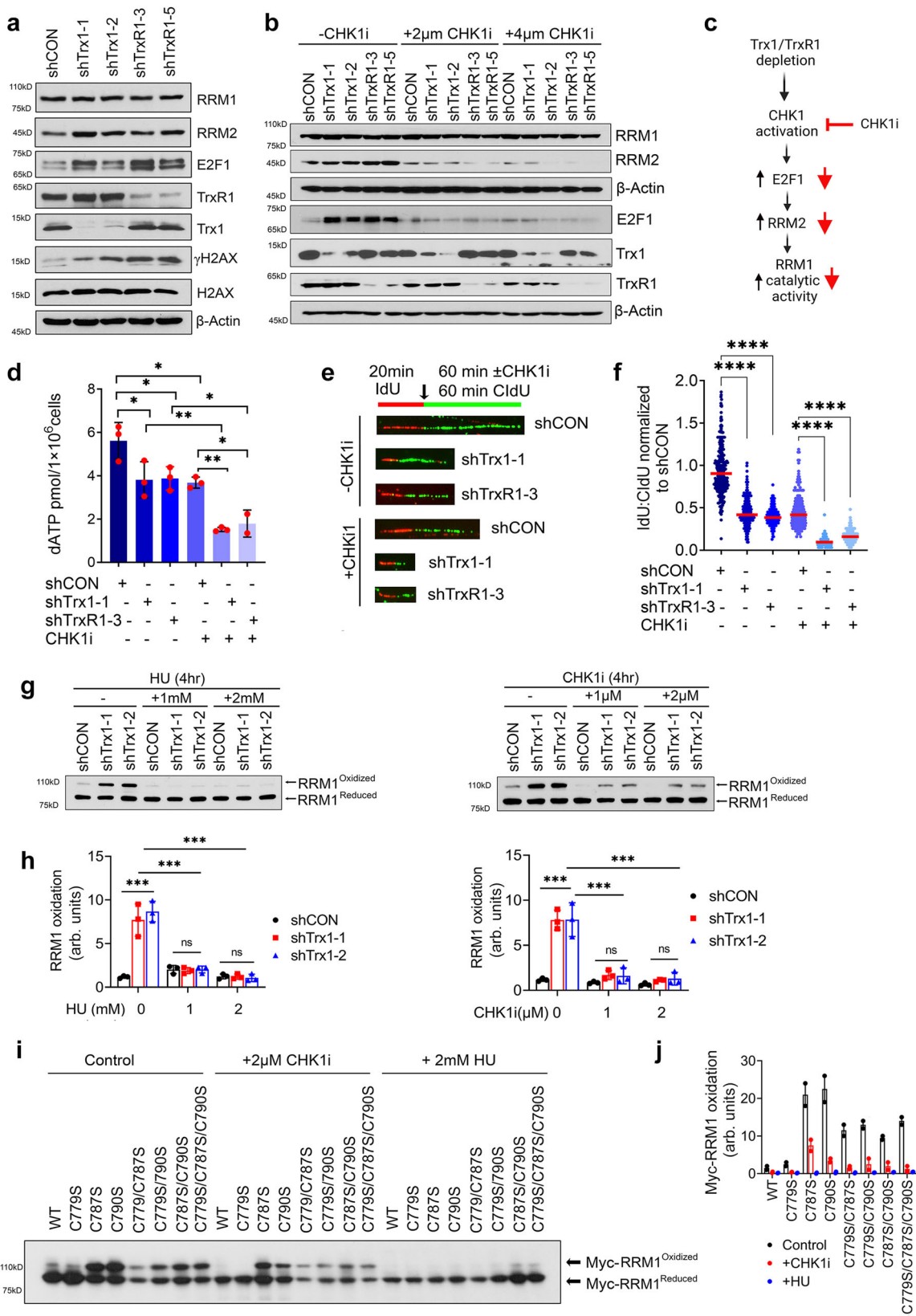

induced by HU-mediated RNR inhibition[44] and that the firing of multiple origins in cells with lower dNTPs increases consumption of dNTPs and fork termination[52,53], we observed an increase in new origin firing and fork termination rate in response to Trx system interruption. Also, the RRM1 oxidation status is closely associated with Trx system interruption-induced RS, which is demonstrated by using a Trx1 -WT

and Trx1- redox mutant expression system. Trx1 contains a thiol-exchange catalytic site at Cys32-X-X-Cys35 and alterations in Cys32 and/or Cys35 can interrupt the flow of electrons due to increased RRM1 oxidation. Evidently, catalytically active Trx1-WT facilitates the recycling of oxidized RRM1 in response to Trx1 loss, resulting in striking mitigation of Trx1 depletion-induced RS and dNTP loss, but redox

**Fig. 7 | Trx1 or TrxR1 depletion triggers the CHK1-E2F1-RRM2 pathway, which can be inhibited by CHK1i. a** Representative western blots of the indicated proteins of the E2F1-RRM1 pathway after Trx1 or TrxR1 depletion. **b** Representative western blots of E2F1 and RRM2 upon Trx1 or TrxR1 depletion and CHK1i treatment at the indicated doses. **c** A schematic diagram *(Created with BioRender.com released under a Creative Commons Attribution-NonCommercial-NoDerivs 4.0 International license)* illustrating the expected impact of CHK1i on Trx1 or TrxR1 depletion-mediated E2F1-RRM2 axis alterations and subsequent RRM1 activity. **d** The levels of dATP in cells with Trx1 or TrxR1 depletion with or without CHK1i treatment ($n = 3$ biological repeats; error bars represent ±SD). Effects on replication fork progression in cells with Trx1 or TrxR1 depletion with or without CHK1i treatment as visualized by track lengths (**e**) and their relative quantification (**f**) ($n = 3$ biological repeats; ≥100 fibers/repeat). Representative western blots of oxidized and reduced RRM1 in Trx1-depleted cells with or without HU or CHK1i treatment (**g**) and their quantification below each set of western blots (**h**) ($n = 3$ biological repeats; error bars represent ±SD). Representative western blots of oxidized and reduced RRM1 in cells expressing various redox mutants of RRM1 treated with CHK1i or HU (**i**) and their quantification (**j**) ($n = 2$ biological repeats). [Statistical information: Data are presented as mean value ± SD. The *p-value*s were calculated using one way ANOVA for multiple comparison. Red line in dot plot (**f**) indicates mean; *≤0.05; **$p ≤ 0.005$; ***$p ≤ 0.001$; ****$p < 0.0001$; ns non-significant].

mutants Trx1(C32S and C35S) were unable to recycle the RRM1 oxidation and/or failed to restore Trx1 depletion-linked phenotypes, such as increased RS and early S phase accumulation, as well as dNTP pool scarcity. The observation that the increased dNTPs accompanied by restoration of replication velocity occurred in Trx1-depleted cells that express catalytically active Trx-WT, but not a redox mutant, suggests that thiol-exchange of Trx1 is critical in regulating enzymatic activity of RNR in mammals. Finally, consistent with the observation in yeast[23], we demonstrated that expression of RRM1-WT restored the Trx1 depletion-induced RS, whereas expression of a RRM1 redox mutant failed to do so. Therefore, our data strongly suggest that RRM1 oxidation contributes to the Trx system interruption-induced RS by causing dNTP deficiency.

The formation of individual cysteine redox-PTMs depends on many factors, including the reactivity of the individual cysteine residue, its surrounding environment and the composition of the local redox environment. A total of 16 cysteine residues are distributed between the N-terminus and the C-terminus of the 792 amino acid-long mammalian RRM1; however, the *E. coli* and *S. cerevisiae* isoforms contain 13 cysteine residues along their 811 amino acid-long chains. As previously reported, two cysteines in the C-terminus of R1/RRM1 (Cys754 and Cys759 in *E. coli* R1 and Cys787 and Cys790 in mammalian RRM1) are critical for the last step of internal electron transfer. The disulfide bond between these two cysteines needs to be reduced by the external redox system to complete the redox and recycling of R1/RRM1. Consistent with the results from bacteria and yeast, we found that the two cysteine (Cys787 and Cys790) are critical for the recycling of RRM1 oxidation in mammalian cells. Our findings using C-terminal cysteine (Cys787 and Cys790) mutants, which fail to undergo thiol-exchange due to Cys→Ser mutation, suggest that thiol exchange at the disulfide bond between Cys787 and Cys790 site is critical for RRM1 oxidation recycling. Indeed, mutations in these two cysteines block the transfer of electrons, as we found an increased oxidized form of RRM1 in cells expressing C787A and/or C790A. In addition, a single mutation at Cys787 or at Cys790 significantly abolishes the RRM1 recycling, indicating that Cys787 and Cys790 are involved in Trx1-mediated RRM1 recycling in mammalian cells. In contrast, a single mutation in another cystine (C779S) proximal to C-terminal Cys787 and Cys790 has no effect on RRM1 recycling. Although it has been suggested that Cys779 in RRM1 is essential for regeneration of RNR activity in vitro in mammalian cells[54], in our study it does not participate in thiol exchange with Trx1 in mammalian cells. It would be very interesting to determine the role of other cysteines in playing specific roles in determining the activity of human RRM1.

Of note, Trx1 or TrxR1 depletion increases ROS, which could cause the oxidized base damage. During the oxidized base repair by base excision repair (BER) pathway, an apurinic site (AP site) is generated and then processed further into DNA single-strand breaks via backbone incision of AP-endonuclease 1. Theoretically, the AP site is processed to DNA single-strand breaks or DSBs when encountered within the replication forks[55]. Thus, we originally speculated that increased ROS contributes to Trx1 or TrxR1 inhibition-induced RS. Surprisingly, ROS is not responsible for the Trx1 or TrxR1 inhibition-induced RS

because the classic ROS scavenger NAC has no significant impact on RS, RRM1 oxidation and antitumor activity as a monotherapy or in combination therapy. Thus, the ROS-induced DNA damage in the condition of Trx1 or TrxR1 interruption are likely rapidly repaired given the fast nature of repair of ROS-induced DNA damage[56,57]. It would be very interesting to determine if Trx1 or TrxR1 inhibition-induced ROS is important for the RS and DNA damage and tumor cell growth suppression in BER-deficient cells

RRM2 has been identified as the regulatory subunit of RNR due to its tyrosyl radical-producing activity, which is important for overall RNR activity. RRM2 transcriptional expression and relocation is regulated by ATR/CHK1 pathway during RS[28,58]. We found that CHK1-mediated E2F1-RRM2 induction was triggered during Trx1 or TrxR1 depletion-induced RS, suggesting a RRM2 expression-associated mechanism may compensate for the dNTP deficiency due to RRM1 oxidation. This compensatory mechanism could funnel the tyrosyl radical to RRM1 to generate sufficient dNTPs to rescue cells stalled in S-phase. In support of our hypothesis, CHK1 inhibition abrogates the Trx1 or TrxR1 depletion-induced RRM2 expression and leads to deceased RRM1 oxidation that depends on the RRM2-mediated tyrosyl radical transfer and more profound dNTP pool scarcity. The additional evidence supporting the loss of tyrosyl radical production capacity of RRM2 following CHK1 inhibition is from the direct measurement of RRM1 oxidation. CHK1 inhibition in Trx1-depleted cells significantly decreases the accumulation of oxidized RRM1, probably because CHK1i-mediated abrogation of RRM2 expression resulted in lower RRM1 oxidation due to decreased tyrosyl activity in Trx1-depleted cells. Further, the accumulation of oxidized RRM1 due to mutation in C-terminal cystines (Cys787 and Cys790) was also abrogated following CHK1i or HU treatment as both mediated the loss of RRM2 activity, leading to the inhibition of long-range electron transfer to RRM1, subsequently leading to abrogating RRM1 oxidation because the RRM2-mediated initial tyrosyl radical transfer is blocked. Thus, our hypothesis is that the disruption of the Trx system induces oxidation of RRM1, a crucial step for the Trx system interruption-induced RS. This occurs by impairing dNTP production due to decreased RNR activity. When CHK1 activity is inhibited in these cells, CHK1 inhibition can further hinder RNR activity by affecting RRM2 through an E2F1-mediated mechanism (Fig. 9m). Therefore, there is a synergistic interaction between Trx1 or TrxR1 inhibition and CHK1i.

The ATR/CHK1 is important for cell survival under conditions of RS via promotion of cell cycle checkpoint, DNA synthesis, replication fork stability and regulation of replication initiation[28]. Thus, we cannot exclude the impact of CHKi-induced cell cycle checkpoint interruption is also important for the synergy because inhibition of Trx system and CHK1 leads to increased catastrophe, a situation where cells try to move to a mitotic stage with the damaged DNA because of the impaired cell cycle checkpoint. Thus, in addition to the DNA synthesis deficiency, abrogation of cell cycle checkpoints might also contribute to CHK1i-induced antitumor activity, especially with Trx system interruption.

Although the most dramatic sensitization has been observed in preclinical studies when CHK1 inhibitors are combined with HU or

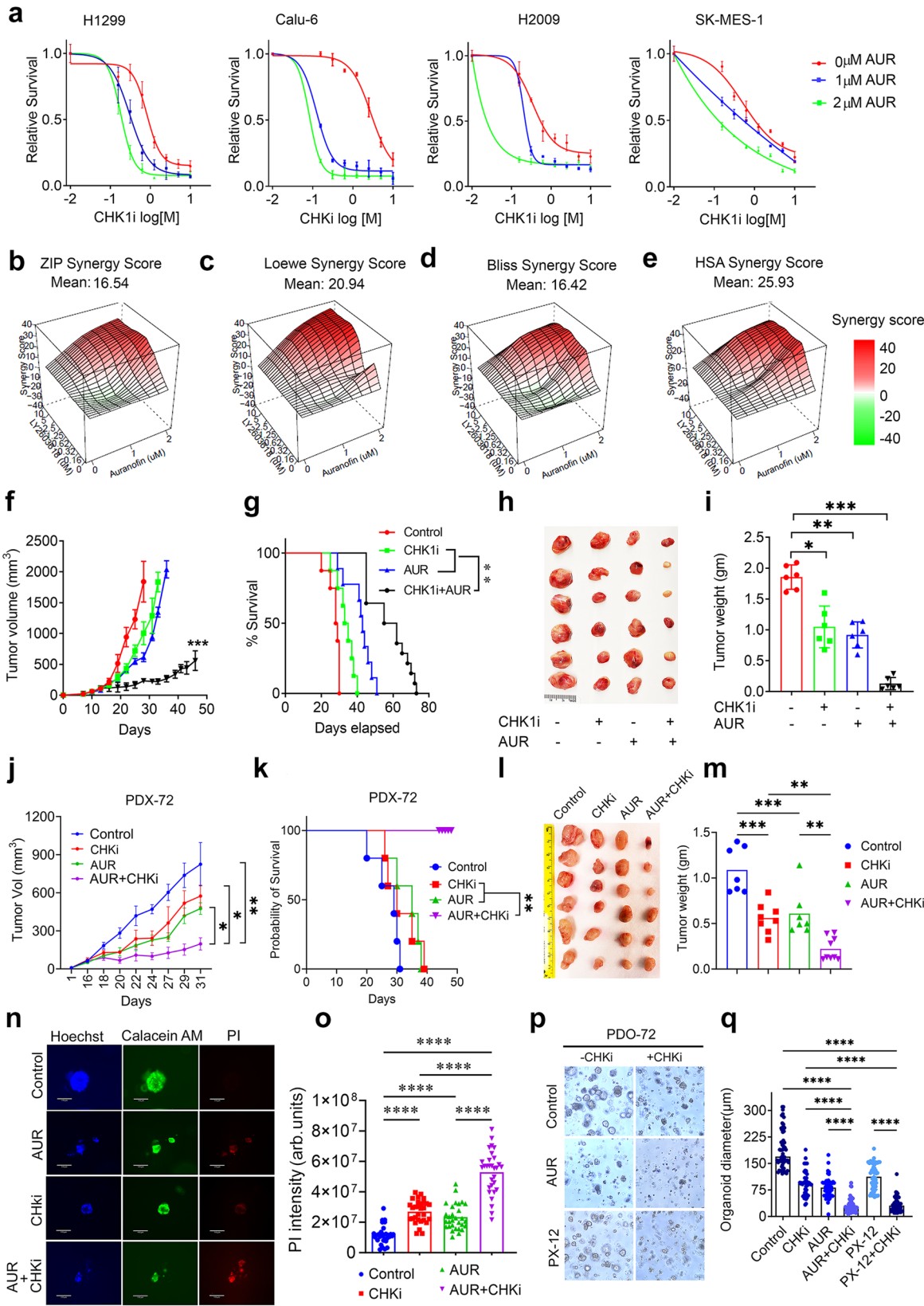

gemcitabine, two antimetabolites that inhibit RNR[2,59], only limited efficacy and toxicity has been noted in clinical trials[5,8,60]. What made us believe that combination of CHK1 and Trx1 or TrxR1 inhibition might be better than the combination with RNR targeting chemotherapy is because of the following three reasons. Firstly, elevated Trx1 or TrxR1 expression in tumor cells cause them to rely on the Trx system to maintain redox homeostasis[61]. In support of the hypothesis that cancer cells are highly dependent on Trx1 or TrxR1 for survival due to their high metabolism and proliferation, loss of the Trx system showed no effect on normal replicative potential. For instance, *Txnrd1*-deficient mouse hepatocytes have equivalent developmental and regenerative proliferative potentials[62]. Indeed, we found that Trx1 or TrxR1

**Fig. 8 | AUR and CHK1i have a synergistic interaction with regard to their antitumor activity and RS. a** Relative survival of different NSCLC cell lines treated with AUR or AUR + CHK1i. ($n = 3$ biological repeats in triplicates; error bars represent ±SD). Different synergy scores, such as ZIP (**b**), Loewe (**c**), Bliss (**d**) and HSA (**e**) indicate the true synergy between CHK1i and AUR. A synergy score ≥10 is considered to be synergistic. **f** Growth rate of the tumors in the indicated groups over the indicated course of time ($n = 10$ animals in each group; error bars represent ±SD). **g** and the survival rates by Kaplan–Meier analysis (Control $n = 8$; AUR $n = 9$; CHK1i $n = 8$; AUR+CHK1i $n = 10$). Representative images of tumors excised from indicated groups (**h**), the average weight of the excised tumors ($n = 6$ tumors form each group; error bars represent ±SD) at the endpoint of treatment (**i**), of a mouse xenograft H1299 model of NSCLC treated with CHK1i (25 mg/Kg) and AUR (10 mg/Kg). **j**–**m** The synergy between AUR and CHKi, as detected by a PDX model (Control $n = 8$; AUR $n = 8$; CHK1i $n = 10$; AUR + CHK1i $n = 8$; error bars represent ±SD). The growth rate of PDX-72 tumors in indicated treatment groups, over the indicated course of time and the survival rates by Kaplan–Meier analysis ($n = 5$ animal/group) (**k**). Representative images of PDX-72 tumors excised from indicated groups (**l**), the average weight of the excised tumors at the endpoint of treatment (Control $n = 7$; AUR $n = 7$; CHK1i $n = 8$; AUR + CHK1i $n = 9$ individual tumor from the indicated groups; error bars represent ±SD) (**m**). **n** Tumor organoid generated from PDX-72 tumors (PDO-72) were stained with Hoechst, Calacein AM and PI to stain nucleus, live cells and dead cells respectively. Representative organoids in indicated treatment groups. **o** The PI intensity, an indicator of cell death of organoids, in each treatment groups ($n \geq 50$ organoids/groups from 2 biological repeats; error bars represent ±SD). **p** Representative images of 10-day-old organoids treated with the indicated inhibitors for 72 h. **q** The growth and size of the organoids from each treatment group ($n \geq 50$ organoids/groups from 2 biological repeats; error bars represent ±SD). [Statistical information: Data are presented as mean value ± SD. The $p$-values were calculated using one way ANOVA for multiple comparison; *$p \leq 0.05$; **$p \leq 0.005$; ***$p \leq 0.001$].

inhibition has much less impact in RS in the untransformed lung cells. Secondly, a very broad deficiency of the Grx-GSH system was observed in human tumors, including NSCLC, which provides a unique opportunity to target a subset of NSCLC by Trx1 or TrxR1 inhibition. It is quite common that loss of the Grx-GSH system due to deficiency in its components and other upstream factors render cancer cells highly reliant on the Trx system and show synthetic lethality with Trx system inhibition[34,63–66]. Lastly, CHK1i, by regulating the RRM2 compensation pathway, provides the foundation for synthetic lethality. Of note, our study also suggests the potential for combining Trx1 or TrxR1 inhibition with homologous recombination (HR) inhibitors as the stalled replication forks triggered by inhibition of RNR activity highly depends on HR-associated repair[44]. It would be quite interesting to determine if Trx1 or TrxR1 inhibition can be combined with HR inhibitors to suppress tumor growth in NSCLC and other human cancers.

We anticipate that the status of glutathione-glutaredoxin (Grx)-GSH system (NADPH, glutathione reductase, GSH and Grx) might impact combinatorial strategies via compensation of Trx system loss[34]. Nevertheless, it remains largely uncertain, particularly in mammals, if these two systems have an overlapping role in promoting RNR redox recycling and dNTP synthesis. Although it has been reported that Trx and Grx-GSH are dithiol electron donors of *E. coli* RR and Grx1 is the most efficient electron donor for the enzyme activity[17], the Trx1 and Grx1 systems show similar catalytic efficiencies as recombinant mouse RR complex[17,67]. Interestingly, a recent study suggested that for human cells, hTrx1 is a much more efficient reductase for RRM1 regeneration than hGrx1[68]. Our study supports the concept that both systems might be important in terms of its regulation of RNR as Trx1 or TrxR1 inhibition has a better sensitization activity in cells with a deficient Grx-GSH system. Thus, investigating a potential a compensatory role for the Trx and Grx-GSH systems in the regulation of RNR activity in mammalian cells is certainly warranted in future studies, including the efficacy of Trx1 or TrxR1 and CHK1 combinatorial inhibition in treating cancer cells with a Grx-GSH deficiency. It is worth noting that other factors may also be important, such as the p53 status as it has been shown that it affects the sensitivity of CHK1 inhibitors. p53 is important for the G1/S checkpoint. Disruption of G1/S checkpoint control due to loss of p53 leaves cells reliant on G2/M arrest for DNA repair when the cells are challenged with DNA damaging agents[2,69,70]. CHK1 phosphorylates and inhibits its substrates, the phosphatases CDC25C and CDC25A, leading to arrest at the G2/M checkpoint[2]. Therefore, p53-deficient cells are normally more sensitive to CHK1 inhibitor-associated cancer therapy. As most of the cell lines in our study harbored *TP53* mutations, it would be interesting to investigate the impact of the p53 status on the efficacy of this combination as well.

The Trx system is suggested to be a promising target for cancer therapy[71]. The anti-rheumatoid arthritis drug AUR is now recognized as a potent TrxR inhibitor, and attempts have been made to repurpose AUR for the treatment of infection and cancers[72–75]. However, the molecular mechanisms underlying the anticancer effects of AUR are not yet completely understood, which limits its choice for a combination therapy. Currently, most studies have focused on ROS and mitochondrial function, including proteasomal inhibition-linked mechanisms, in the mechanisms underlying the anti-tumor effects of AUR[65,76–78]. In human lung cancer, AUR induces cytotoxicity via ROS production in stem cell-like cancer cells[75]. The primary molecular targets of AUR have also been described to be mitochondrial and (to a lesser extent) cytoplasmic TrxR (TrxR1)[76]. The anticancer activity and anti-rheumatoid impact of AUR have been found to be linked to inducible ROS and/or glycolysis[75,77,79–82]. In our study, AUR has a synergistic interaction with CHKi and AUR treatment increased RS due to poor dNTP production, resulting in defective replication fork elongation following accumulation of oxidized RRM1. Here, we have shown that AUR treatment leads to RS and DNA damage and identify an unexplored mechanism of redox inhibition of RRM1 consequently causing RS. Thus, Trx system inhibition-induced deficiency most likely contributes to RS and antitumor growth activity. In the future, the challenge will be to identify the determining factors for the antitumor activity of AUR and any different mechanisms that explain these effects, as well as identifying the patient population that would most benefit from its treatment.

Our study provides mechanisms by which AUR in combination with CHK1i inhibits cancer cell growth via regulation of cysteine oxidation of RRM1. Thus, repurposing of this drug in combination with CHK1 inhibitors for NSCLC treatment might be an area to focus on. Although AUR is considered safe for human use in treating rheumatoid arthritis and the drug has a well-known toxicity profile[83], the dose used for its antitumor activity in current preclinical studies, including here, is higher compared to those used in clinical settings for the treatment of rheumatoid arthritis. The next clinical steps could be validation of AUR-induced CHK1i sensitization in NSCLC or other tumor models in clinical settings to study the antitumor activity in an immune component host. In addition, more specific and potent inhibitors targeting Trx1 or TrxR1 needs to be developed.

In summary, the interruption of Trx1 or TrxR1 increases RS due to depletion of the dNTP pool by inhibition of RNR redox recycling. We propose that increased RRM1 oxidation following Trx system inhibition and the abrogation of RRM2 pathway by CHKi contributes to the synergistic interaction by the severe loss of RNR function. Our study reveals an underappreciated mechanism responsible for the Trx or TrxR1 inhibition-induced RS, and we propose a potential combinational approach to treat a subset of NSCLC. We foresee great potential for future efforts towards developing inhibitors for targeting the Trx system as a monotherapy or in combination with other targeted therapies, from a redox perspective involving the Trx system.

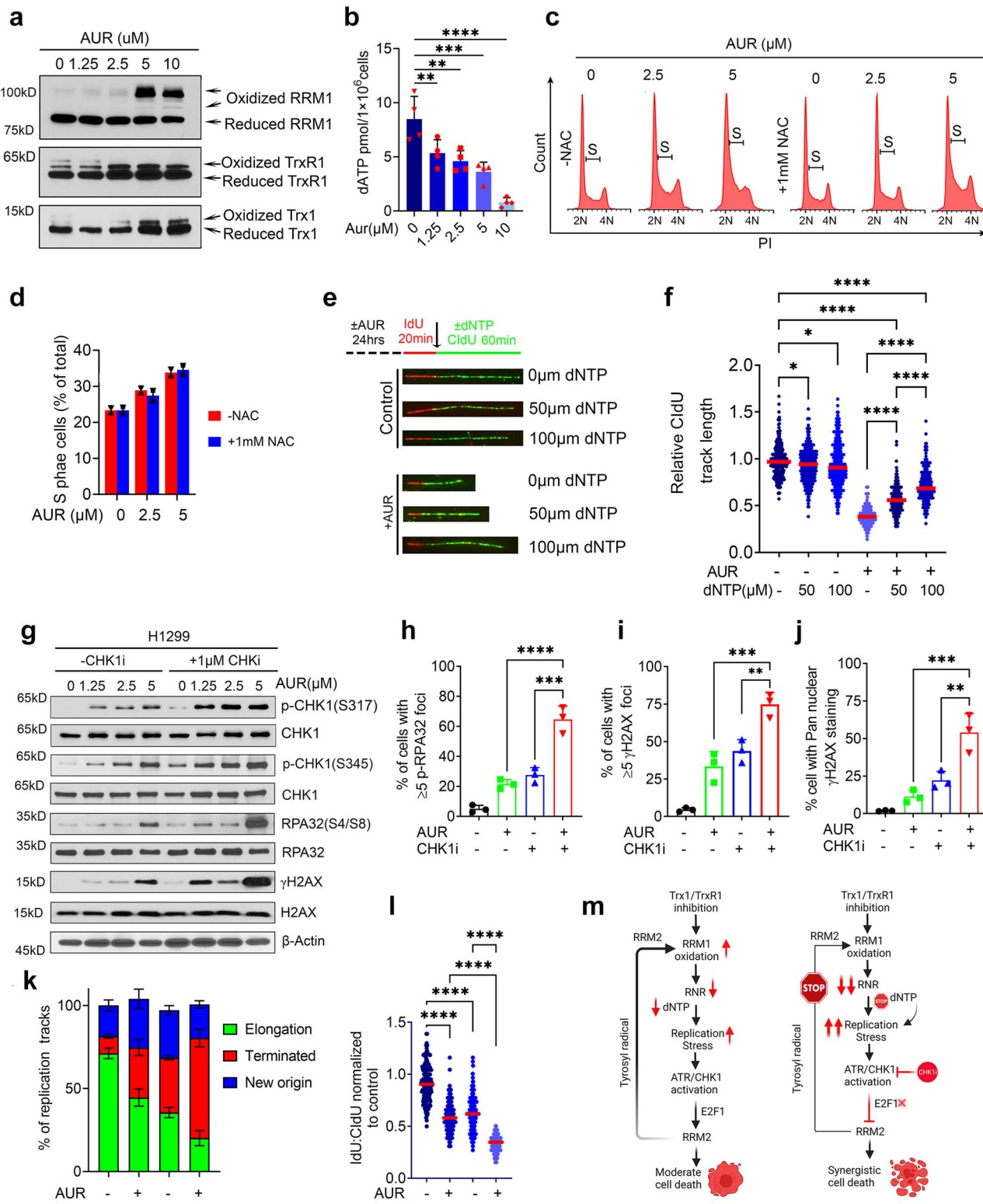

## Methods

### Cell line, cell culture reagents and inhibitors

All experiments complied with protocols and conditions approved by the Office of Research and Institutional Animal Care and Use Committee of The Ohio State University (Columbus, OH). The NSCLC cell lines H1299, A549, Calu-6, SK-MES-1, H2006, H460 and H1437 were purchased from ATCC and maintained in DMEM (H1299 and A549); MEM (Calu-6 and SK-MES-1) or RPMI1640 (H2006, H460 and H1437) medium supplemented with 10% FBS and 1% penicillin/streptomycin. Cell lines were routinely screened for mycoplasma contamination using LookOut® Mycoplasma PCR detection kit from Sigma (Cat. MP0035). All cell lines were authenticated via STR profiling by Genomics Shared Resources at the Ohio State University. Cells were cultured at 37 °C in a humidified incubator at 95% humidified oxygen and 5% CO2. The CHK1 inhibitor LY2603618 was purchased from APExBIO (cat. #A8638) and the ATR inhibitor VE-821

**Fig. 9 | AUR treatment increases RRM1 oxidation and impairs replication fork elongation due to depletion of the dNTP pool. a** Representative western blots of oxidized and reduced RRM1 in H1299 cells treated with AUR. **b** Levels of dATP in the AUR-treated cells ($n = 4$ biological repeats; error bars represent ±SD). Effects of NAC on the accumulation of S phase cells caused by AUR treatment (**c**) and its quantification ($n = 2$ biological repeats; error bars represent ±SD) (**d**). FACS gating strategies are presented in Supplementary Fig. S17. Effects of AUR treatment on replication fork progression with or without dNTP supplementation as visualized by DNA fiber tracks (**e**) and their relative quantification (**f**) ($n = 3$ biological repeats; ≥100 fibers/repeat). **g** Representative western blots of RS markers in cells treated with AUR and an CHK1i. Percent of cells with p-RPA32(S33)-positive foci (**h**), those with ≥5 foci of γH2AX (**i**) and with pan-γH2AX nuclear staining (**j**) in those treated with AUR, a CHK1i or both ($n ≥ 100$ cells/repeat from 3 biological repeats; error bars represent ±SD). **k** The effect of combined AUR and CHK1 inhibition or

monotherapy on the percentage of replication fork progression, elongation termination and new origin firing ($n = 3$ biological repeats; ≥100 fibers/repeat; error bars represent ±SD). **l** The effects of AUR and CHK1 inhibition or monotherapy on replication fork progression. The dot plots show the relative CIdU track length ($n = 3$ biological repeats; ≥100 fibers/repeat). **m** A schematic diagram *(Created with BioRender.com released under a Creative Commons Attribution-NonCommercial-NoDerivs 4.0 International license)* illustrating our working model of how Trx1/TrxR1 depletion or inhibition synergistically interacts with CHK1i to promote more anti-tumor effects than either therapy alone. In this model, both RRM1 and RRM2 are involved in the synergistic interaction between Trx system inhibition and CHK1i. [Statistical information: Data are presented as mean value ± SD. The *p-value*s were calculated using one way ANOVA for multiple comparison. Red line in dot plot (**f** and **l**) indicates mean; *$p ≤ 0.05$; ** $p ≤ 0.005$; *** $p ≤ 0.001$; **** $p ≤ 0.0001$; ns: non-significant].

(cat. #S8007) was purchased from Selleckchem. Auranofin (Cat. #A6733) and N-Acetyl cysteine (cat. #A9165) and Iodoacetamide (IAA, cat. #A3221) were purchased from Sigma Aldrich.

## Genome-wide lentiviral shRNA screening

A human whole-genome library (RHS6083, GE Dharmacon) was transfected into the NSCLC cell line H1299 to conduct the genome-wide lentiviral shRNA screening[84]. The screening procedure was conducted according to the manufacturer's instructions. 1 μmol/L of LY2603618 was used for CHK1 inhibition. Genomic DNA extraction, PCR amplification of shRNA and PCR product purification were performed according to the protocol provide by RHS6083. Purified amplicons from different samples were pooled and analyzed with a Bioanalyzer (Agilent) and qPCR and then sequenced on an Illumina HiSeq 2500 platform on high output mode. TrimGalore was used for trimming and quality evaluation of the sequences from the shRNA samples. Sequences that passed the quality filters were aligned to the shRNA library sequences provided by Dharmacon (GE). The R package DESeq was used to determine whether there was differential expression. An initial significance cutoff was applied to the DESeq output, and only shRNAs that had an adjusted $P < 0.05$ false discovery rate (FDR) were considered.

## Immunohistochemistry (IHC)

A tissue microarray (TMA) with 50 NSCLC samples and their matched adjacent normal tissue (ANT) were purchased from TissueArray (slides ID: LC10013c #199 and LC10019c #200) and slides were subjected to graded alcohol following deparaffinization. Monoclonal Trx1 (1:100; cell signaling #2429 clone-C63C6) and TrxR1 (1:50; #15140 clone-D1T3D) antibodies were used for IHC. The median immunoreactivity score (IRS) was determined from the staining intensity (SI) and percentage of positive cells (PP): IRS = SI × PP. An IRS ≥ 1–3 score was classified as positive.

## Plasmids and lentiviral preparation

Plasmids pCMV ΔR8.2 (Addgene ID-12263) and pCMV-VSVG (Addgene ID-8454) for lentiviral packaging were procured from Addgene. Flag-Trx1 WT (Addgene ID-21283); Flag-Trx1 C32S (Addgene ID-21284); Flag-Trx1 C35S (Addgene ID-21285) were procured from Addgene. RRM1 expressing plasmids (Myc-RRM1 WT; Myc-RRM1 C779S; Myc-RRM1 C787S; Myc-RRM1 C790S; Myc-RRM1 C779S/C787S; Myc-RRM1 C779S/C790S and Myc-RRM1C779S/C787S/C790S) were described previously[68]. The validated shRNAs construct of Trx1 (TRCN0000064278, TRCN0000064279) and TrxR1 (TRCN0000046533, TRCN0000046534) were purchased from Sigma-Aldrich and the lentivirus were produced using CaCl₂ mediated transfection into 293T cells. 3'UTR targeting Trx1 shRNAs (3'UTR shTrx1-1; clone ID-V2LHS_275263 and shTrx1-2; clone ID- V3LHS_412949 used in Fig. 6C–H) were purchased from Dharmacon.

## Western blot

Immunoblotting was conducted as previously described[85–87]. Briefly, total protein was extracted using protein extraction buffer (62 mM Tris-HCl pH 6.8; 2% SDS; 10% Glycerol and 10 mM DTT) supplemented with protease inhibitor cocktail (Roche #11836153001) and phosphatase inhibitors (Sigma-Aldrich #P2850) 1 mM PMSF (Sigma-Aldrich #93482). A total of 50 μg protein was separated on denaturing SDS-PAGE and transferred on to PVDG membrane. Non-specific biding was blocked using 5%BSA in TBST for 2 hrs at RT. Proteins of interest were detected using appropriate primary antibodies. Antibodies [(Trx1; #2429 clone-C63C6; TrxR1; #15140 clone-D1T3D; p-CHK1 (S345); #12302 clone-D12H3; p-CHK1 (S317); #8191 clone-D7H2; CHK1; #2360 clone- 2G1D5; p-ATR(T1989); #30632 clone-D5K8W; ATR; #13934 clone-E1S3S; Histone H3; #4499 clone-D1H2; Histone H2AX; #7631 clone-D17A3; RPA32; #2208 clone-4E4 and RRM1; #8637 clone-D12F12; RRM2; #65939 clone-E7Y9J; E2F1; #3742, cleaved PARP1 #5625 clone-D64E10, Cleaved caspase 3; #9661 clone-5A1E); antibodies were used at 1:1000 dilution] were purchased from Cell Signaling Technologies. The anti-p53R2 antibody was purchased from Santa Cruze Biotechnology (#sc-137174 clone A-5; 1:1500). The anti-β-actin antibody was purchased form Sigma Aldrich cat#A5441. Antibodies to detect p-RPA32(S4/8) (cat# A300-245A) and p-RPA32(S33) (cat# A300-246A) were purchased from Bethyl Laboratories. The anti-γH2AX antibody (cat#05-636; clone-JBW301; 1:2000) was procured from EMD Millipore. Anti-p-RPA32 (S33 and S4/8) antibodies were used at 1:1500 and anti-β-actin was used at 1:5000 dilution. All other primary antibodies were use at 1:1000 dilution in 5% BSA in TBST at 4 °C overnight with gentle shaking. After incubation membranes were washed 3 time with 1X TBST for 5 min. The membrane was incubated with relevant HRP-tagged secondary antibodies (Anti-rabbit IgG (#7074); Anti-mouse IgG (#7076) and Anti-rat IgG (#7077) from Cell signaling technology) at 1:5000 dilution in 5% BSA at RT for 2 h with gentle shaking. After washing with TBST, the signal was detected using SuperSignal™ West Pico PLUS Luminol reagent (Thermo Scientific #34578) on X-ray films. Full scan of blots is present in source data and supplementary data file.

## Cell proliferation and cellular toxicity assay

NSCLC cell lines were seeded in 96 well plates and treated with the desired concentration of inhibitors at the indicated durations. After completion of drug treatments, cells were incubated for an additional 3 h in 6 mg/ml MTT solution in growth medium. Cell viability was assessed by absorption measurement of MTT crystals after solubilizing into MTT solvent.

## Annexin V/PI apoptosis assay

Apoptosis in cultured cells was measured using an eBioscience™ Annexin V Apoptosis Detection Kit (Invitrogen; #Cat- 88-8102-72) according to the manufacturer's protocol.

## Xenograft studies and the PDX model

Athymic male and female mice (Strain code: 553, NCI Frederick) at 4–6-week-old were used in this study. Animals were bred at The Ohio State University (Columbus, OH). All mice were maintained under barrier conditions, and the experiments were conducted using protocols and conditions approved by the Institutional Animal Care and Use Committee of The Ohio State University (Columbus, OH). Xenografts were established by injections of H1299 cells ($0.5 \times 10^6$ cells) subcutaneously into flanks of the animals. Tumor diameters were measured with digital calipers, and the tumor volume in $mm^3$ was calculated using the formula: Volume = (width)$^2$ × length/2. Once tumor volume reached 100–150 $mm^3$, the mice were treated with vehicle control or a CHK1 inhibitor (25 mg/kg of LY2603618) via intraperitoneal injection (IP) twice a day for 3 days, followed by 4 days of rest over the course of 3 weeks. In case of combined CHK1i and AUR, AUR was given via IP at 10 mg/kg for 5 days every week accompanied by the same CHK1i regimen. PDX-72 was used to measure the effect of the efficacy of drug combination[49]. PDX72 was established by transplanting freshly removed NSCLC tumor tissue (KRAS G12C mutation) into NSG mice subcutaneously[49]. Anesthetized NSG (NOD *scid* gamma (NSG™)) mice were inoculated with small pieces of PDX-72 tumors. Animals were treated with buprenorphine at 0.1 mg/kg for 3 days for pain management. All mice were maintained under barrier conditions, and the experiments were conducted using protocols and conditions approved by the Institutional Animal Care and Use Committee of The Ohio State University (Columbus, OH). Tumor size ≈2000 $mm^3$ considered as the end point for all experiments and no tumor exceeded the size in this study. At the end point, animals were sacrificed, tumors were excised, weighed and photographed.

## Immunofluorescence staining and intensity profiling of individual cells

Immunofluorescence assays were performed as described previously[85,87,88]. Images were captured using confocal microscope at 20× and 63×. Higher magnification images were used to count the number of foci/cells. 20× were used to measure the intensity of staining in each cell using ImageJ. Briefly, images were first converted into 8-bit grayscale and the threshold was set to visualize the DAPI area and nuclear intensity of RPA32, p-RPA32 (S33) and γH2A was measured. Intensity of each nucleus (at least 1500–2000 cells/group) was recorded, and the data was transferred into GraphPad prism for statistical analysis.

## Neutral comet assay

The Neutral Comet Assay was performed using the Comet Assay kit (4250-050-K) from Trevigen, following the manufacturer's instructions. The length of comet tail was measured using Image J.

## Measurement of ROS by DCFDA

Intracellular ROS was measured by staining of cells with DCFDA under the desired treatment conditions. Briefly, cells were grown in multiwell plates and treated as indicated and harvested through trypsinization. Cells were washed in cold 1× PBS and stained with 10 μM DCFDA in the presence or absence of NAC for 30 min at 37 °C. DCFDA was then washed out and cells were resuspended in 1× PBS before detection of fluorescent signal. ROS assay flow cytometric histograms were generated and analyzed on an BD Fortessa instrument using FACSDIVA™ software (BD Biosciences). DCFDA measurement was also assessed via microplate reader by measuring the DCFDA fluorescence at Ex/Em: ~492–495/517–527 nm.

## Cell cycle analysis

Cells were grown for 2–3 generations in multiwall plates and logarithmically growing cells were harvested by trypsinization and fixed in 70% methanol at −20 °C overnight. After washing in 1× PSB twice cells were resuspended in 1× PSB containing 50 μg/ml PI & 100 μg/ml RNase A and incubated at 37 °C for 1 h before analyzing on an BD Fortessa instrument using FACSDIVA™ software (BD Biosciences).

## BrdU incorporation assay

BrdU incorporation assay was performed using PhaseFlow™ FITC BrdU kit according to the manufacturer's instructions. Briefly, cells were pulsed with 10 μM BrdU or 30 min and stained with anti-BrdU antibody following DNA digestion. Cells were then stained with nuclear stain 7-AAD, and data was recorded using an BD Fortessa instrument.

## DNA fiber assay

DNA fiber assays were performed as previously described[84,85,88]. Briefly, cells were pulsed with 50 μM IdU for the indicated durations. IdU was washed out with pre-warmed 1× PBS and 200 μM CIdU was then added for desired duration with or without inhibitor or dNTPs. Cells were then harvested and lysed on a slide and DNA fibers were stretched via a drop rolling method. Fibers were dried and fixed in 1:3 acetomethenol solution for 10 mis. Fibers were denatured in 2.5 mM HCl and rehydrated in 70% ethanol followed by nonspecific blocking in 2% BSA + 1% goat serum for 30 min at RT. BrdU antibody (#347580, 1:20, BD Biosciences) and CldU antibody (ab6326, 1:200, Abcam) were incubated overnight at 4 °C. Next day, fibers were washed and appropriate 2° antibodies (Goat anti mouse Alexa fluor 594; Cat# A-11006 and Goat anti Rat Alexa Fluor 488; Cat# A-11006 at 1:400 dilution) were incubated for 1 h at RT in dark. Fibers were mounted into antifade mounting medium. Fibers were imaged at 40× and fibers length was measured using Image J. Aa total of 100–200 fibers per condition was counted and experiment was repeated 3 time independently in double blind fashion.

## BrdU incorporation assay

BrdU incorporation assay was performed using PhaseFlow™ BrdU kit from Bio Legend (cat. 370704) according to manufacturer's protocol.

## dNTPs measurement

The dNTP measurement was conducted as previously described[89]. Briefly, cell pellets ($5 \times 10^4 \sim 2 \times 10^6$ cells) were washed twice with 1× DPBS (Mediatech, VA), and resuspended in 100 μl of ice-cold 60% methanol. Samples were vortexed vigorously to lyse the cells and then heated at 95 °C for 3 min, prior to centrifugation at $12,000 \times g$ for 30 s. The supernatants were collected and completely dried under vacuum, using a SpeedVac (Savant, NY) with medium heat. The dried pellets were subsequently resuspended in dNTP buffer (50 mM Tris-HCl, pH 8.0 and 10 mM MgCl2; 100 μl for $1 \times 10^6$ HeLa cells, and 10 μl for $1 \times 10^6$ primary cells) and usually 1-2 μl of the extracted dNTP samples were used for each 20 μl single nucleotide incorporation reaction. The proper dilutions of the dNTP samples were prepared for the assay to make the primer extension values lie within the linear ranges of the dNTP incorporation (2-32% primer extension). The extracted dNTP samples were stored at −70 until used. Several different volumes of the extracted dNTP samples were also used to confirm the linearity of the primer extension. In addition, the dNTP samples were prepared from different cell numbers, depending on the recovery efficiency of the primary cells from each blood sample (see below). However, the dNTP content of each cell type was normalized by pmole/$1 \times 10^6$.

## Redox western blot

Redox western blots were performed as previously described[47]. Briefly, cells were harvested and pelleted by addition of 100% TCA to final concentration of 10% and pellet was washed twice in acetone. Alkylation of free thiols was achieved by resuspending the TCA extracted pellet into alkylation solution containing 75 mM iodoacetamide, 1% SDS, 100 mM Tris·HCl (pH 8), 1 mM EDTA following incubation at 25 °C for 15 min. Further, alkylated sample was

dephosphorylated using calf intestinal phosphatase (CIP) for 1 h at 37 °C. Dephosphorylated-alkylated protein samples were subjected to non-denaturing SDS-PAGE (-DTT; no boiling) or denaturing (+DTT and boiling) as indicated.

## Synergy score
Synergy scores were generated using online SynergyFinder 2.0 web application.

## Organoid culture
PDX-72 untreated tumors were dissociated using a human tumor dissociation kit (Miltenyi Biotech; Cat# 130-095-929) following the manufacturer's protocol. Dissociated tissue was filtered via a 70 µm filter and subjected to isolation of human tumor cells using a human cancer cell isolation kit (Miltenyi Biotech; Cat# 130-108-339). Isolated tumor cells were then subjected to depletion of the mouse cell contamination using a Mouse cell depletion kit (Miltenyi Biotech; Cat# 130-104-694). The purity of the human cancer cells was verified using anti-human-EpCAM (CD326) from Miltenyi Biotech (Cat3130-111-000; clone REA764; 1:100 dilution). These tumor cells were then imbedded in matrigel (Corning; Cat#356231) and cultured in organoid medium according to a previously published protocol[90]. The diameter of the organoids was measured with Image J. To detect the degree of cell death in organoids, organoid medium was supplemented with Hoechst (10 µg/ml) (Invitrogen, Cat#H21486), Calcein AM (5 µM) (Invitrogen; cat#C3100MP) and PI (10 µg/ml) (Invitrogen; Cat#P1304MP) and incubated for 1 h under the growth condition. Organoids were then imaged on an Echo Revolve fluorescence microscope. The PI intensity of each organoid was measured with Image J.

## Statistical analysis and reproducibility
Results are depicted as mean ± SD unless indicated otherwise. GraphPad Prism 9.0 software (La Jolla, CA) was used for statistical analysis as described within figure legends. $P$-value ≤ 0.05 was considered statistically significant. DNA fiber assay and dNTP assays were performed in double blinded fashion. Animals were randomly divided in different groups before the initiation of the drug injections. Animals with visible occurrence of ulcer were excluded from the studies. All experiments were repeated at least 3 times independently with similar results unless indicated otherwise. One way ANOVA was used to compare multiple groups without any adjustment. Statistical analysis details are provided in the end of each figure legends to save the space; $p$ values are indicated with * (*$p$ ≤ 0.05; **$p$ ≤ 0.005; ***$p$ ≤ 0.001; ****$p$ ≤ 0.0001; ns: non-significant*].

## Reporting summary
Further information on research design is available in the Nature Portfolio Reporting Summary linked to this article.

# Data availability
The authors declare that all the other data supporting the findings of this study are available within the article and its supplementary information including source data files. Genome-wide lentiviral shRNA sequencing data is uploaded to GEO (accession ID-GSE263176) which is publicly available to readers. Source data are provided with this paper.

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

## Acknowledgements

We thank Dr. Jimin Shao (Zhejiang University Cancer Center, Zhejiang University School of Medicine, China) for providing the valuable RRM1 expression plasmids; The work described herein was supported by grants (R01 CA249198), and the Lung Cancer Discovery Award, and DOD LCRP, W81XWH2010868 and Pelotonia Idea Award to J.Z. from the National Cancer Institute and American Lung Association and U.S. Department of Defense and the Ohio State University James Comprehensive Cancer Intramural Research Program respectively; This research was also supported by the seed grand of Department Radiation Oncology, The Ohio State University to C.B.P.; The project was also supported by the National Center for Advancing Translational Sciences (grant no. UL1TR002733). dNTP data was generated using the funds from the grants NIH R01 AI136581 and NIH R01 AI150451 to B.K. Research reported in this publication was supported by the Ohio State University Comprehensive Cancer Center (OSUCCC) and the National Institute of Health under grant number P30 CA016058. We thank the Target Validation Shared Resource (TVSR) at the OSUCCC for providing the NSG mice used in the preclinical studies described herein. The content is solely the responsibility of the authors and does not necessarily represent the official views of the National Institute of Health. We also acknowledge the Flow cytometry shared resources (FCSR) and Genomic shared resources (GSR) at the OSUCCC for flow cytometry and sequencing services respectively. We also acknowledge Comparative Pathology & Digital Imaging Shared Resource at OSUCCC for immunohistochemistry services. Artwork prepared in Figs. 1a, 9m and 7c and Supplementary Figs. S4a, S5c, and S13a was created with BioRender.com with a valid institutional license from BioRender through The Ohio State University.

## Author contributions

C.B. Prasad: Conceptualization, data curation, statistical analysis, methodology, writing–original draft, writing–review and editing. A. Oo: dNTP data generation. Y. Liu: data curation and methodology. Z. Qiu: Genome wide screening and methodology including draft editing, Y. Zhong: methodology including PDX model. Na Li: Methodology and editing. D. Singh: Methodology and draft editing. X. Xin: Methodology and editing. Y. Cho: Methodology. Z. Li: Methodology. X. Zhang: TCGA data curation, writing–review and editing. C. Yan: Methodology and editing. Q. Zheng: Review and editing. Q. Wang: Methodology and editing. D. Guo: Methodology. B. Kim: dNTP Methodology. J. Zhang: Conceptualization, funding acquisition, investigation, writing–review and editing.

## Competing interests

The authors declare no competing interests.
