## [Peer Review File · Nature Communications]

The thioredoxin system determines CHK1 inhibitor sensitivity via redox-mediated regulation of ribonucleotide reductase activityEditorial Note: Parts of this Peer Review File have been redacted as indicated to maintain the confidentiality of unpublished data.

REVIEWER COMMENTS

Reviewer #1 (Remarks to the Author):

In this study, Prasad and colleagues investigate the mechanistic basis of thioredoxin-1/TrxR1-based modulation of Chk1i vulnerability in NSCLC lines. The authors present an interesting and physiologically valid rationale for why inhibiting the thioredoxin system enhances Chk1i-induced replication stress. This investigation goes well beyond the obvious attribution of the observed phenotype to ROS detoxification systems modulated by Trx1 function, adding significant mechanistic depth to this study. The authors propose this co-inhibition relies on dysregulation of reductive biosynthesis by ribonucleotide reductase, leading to depletion of the deoxyribonucleotide pool and replicative stress-induced genotoxic damage that is independent of ROS. The overall experimental strategy for this study is extremely well-designed, with solid consideration of the key mechanistic factors needed to support their hypothesis and with appropriate methodical rigor. In particular, there is a great deal of elegant data provided in Figs. 5 and 6 supporting involvement of replication stress, the role of a depleted nucleotide pool and RRM oxidation. However, there is some unevenness in interpretation of data and some gaps in characterization. Specific concerns are described below. Experimental and/or textual clarification provided by the authors regarding these points would significantly strengthen their innovative study.

- Fig 2: The reduction in tumor weight and volume are very impressive. The authors are encouraged to also provide H&E images to show the overall changes in cellularity under the specific treatments. Further characterization of tumor tissues could add mechanistic heft to these *in vivo* experiments and make them consistent with the extensive *in vitro* studies. For example, *in vitro*, the H1299 cells presumably undergo cell death (which would contribute to a durable treatment response as opposed to growth arrest). It would be useful to know if these significantly reduced tumor burdens are also due to induction of apoptosis. The authors should ideally provide TUNEL images or tumor western blots showing cleaved PARP or cleaved caspase levels in support of this information. Additionally, how are DNA damage markers/replication stress markers altered across the control and differentially treated tumors?
- Fig 2: there is some variability in the tumor sizes although only the smallest are shown in Fig 2F. If the levels of Trx1 or TrxR1 knockdown are evaluated across the Chk1i treated tumors, do the smaller tumors have better knockdown compared to those tumors that have grown bigger? These data would directly support that co-inhibition of the Trx1 system cooperates with Chk1i-induced tumor suppression *in vivo*.
- Regarding the patient-derived data analysis in Fig 1, it should be noted is that H1299 is NRAS-driven, which is not typical of human NSCLC. The Calu line is driven by KRAS G12C, the more human-representative oncogenic driver of NSCLC. KRAS-driven tumors are enriched for the thiol redox-protective interactome; however, this is not known for NRAS. To extend the disease-physiologic relevance of their *in vivo* results which are in the H1299 line, authors should ideally address this by specifically evaluating public patient datasets for whether the small subset of NRAS driven LUAD also exhibit elevated Trx1, TrxR1, or if there is correlation between oncogenic NRAS and dependency on this interactome.

- Fig 3. With the extensive degree of DNA breaks produced by this treatment, TUNEL should be supplemented by immunoblotting for molecular markers of cell death to support the cytotoxicity evoked by combinatorial treatment. If these are not seen, the S-phase arrest observed may indicate senescence-like arrest as the predominant tumor suppressor response. This distinction also impinges on my comments for Fig. 2 above. Cell death-inducing treatments in general engender a more durable anti-tumor response. If this combination is indeed evoking cell death, then this point could be reliably made in the discussion.

- Are these cells wt for p53 or p16? If so, these markers should also be evaluated as these factors will impinge on how pervasively these treatments can be translated to patients. At the very least, a note should be included in Discussion regarding the dependency of these treatments on key tumor-suppressor pathways, based on their status in the cell lines used in this study.

- Fig 4: The DCFDA flow profiles should be quantified, with appropriate statistics added. More importantly, the authors should be aware that demonstrating NAc-mediated rescue of DCFDA-detected ROS levels is no longer considered best-practice to indicate ROS-mediated phenomenon. NAc decreases DCFDA signal by reducing the fluorophore itself rather than detoxifying cellular ROS (which is chemically not possible for NAc). Thus, the lack of effect of NAc on the other cellular outcomes from Trx1 or TrxR1 inhibition does not definitively rule out ROS involvement by virtue of its effect on the DCFDA profiles. In this regard, the authors should use a direct ROS scavenger to make their point and/or culture cells at lower ambient oxygen.

The striking ROS (total peroxide) induction observed upon both TRX1 and TRXR1 depletion, coupled with the observed DNA breaks, almost certainly suggest some ROS involvement. These peroxides are labile and easily interconverted into other species such as hydroxyl radicals which can damage DNA and the nucleotide pool. Thus, the observed S-phase arrest, cytotoxicity and genotoxic damage is very likely the aggregate result of a depleted nucleotide pool and direct generation of breaks/nicks, which would account for the punctate and persistent DSB foci. This would not detract from the conclusions made by the authors and could add additional mechanistic nuance should other methods of ROS rescue not provide congruent results as NAc.

- To fully evaluate the role of ROS (or lack thereof) in the observed phenomena, the authors are encouraged to evaluate 8-oxoguanine and/or lipid peroxide levels, as these may be unappreciated contributors to the observed cytotoxicity of the dual treatment. At the very least, these parameters could help rule out or define the role played by ROS in the dual treatment settings.

- Auranofin is used in these studies to chemically inhibit TrxR1 in addition to genetic knockdown, which is appropriate and rigorous. Although this is intended to support the lack of Trx1 site reductive regeneration by TrxR1, which is critical for RRM function, TrxR1 inhibition can affect cancer cell proliferation and survival through Trx1-independent mechanisms (i.e. by inhibition of protein tyrosine phosphatase 1B function). As Trx1 directly influences RRM function, the authors are encouraged to repeat the in vitro experiments with the Trx1 specific inhibitor, PX-12. The outcomes would likely be informative either way about the mechanisms by which the thioredoxin interactome synergizes with Chk1 inhibition. If the Px-12 results do not directly recapitulate auranofin, there may be a signaling component (through Ask1 or ATM/ATR downstream signaling) that controls the response to Chk1i

inhibition, in addition to replicative stress.

- Textual point – please label immunoblots with molecular weight markers for the various signals provided

Reviewer #2 (Remarks to the Author):

CHK1 inhibition (CHK1i) adjuvant therapy has shown promising efficacy in preclinical models but only minimal efficacy with substantial toxicity in clinical trials. To identify novel strategies to enhance CHK1i efficacy, Prasad et al. have identified the master redox regulator Trx1 as a new determinant of CHK1i sensitivity through a high throughput screening. The authors showed that Trx1 knockdown/inhibition combined with CHK1i had a synergistic effect potentially via triggering synthetic lethality in NSCLC. They further found that Trx1 knockdown in combination with CHK1i promoted profound replication stress (RS) and cell death. Mechanistically, they found that Trx1 knockdown/inhibition abrogated dNTP production and impaired replication fork elongation to induce RS mainly through increasing the oxidation of ribonucleotide reductase (RNR) subunit RRM1, but not ROS. Moreover, CHK1i treatment disrupted the Trx1 depletion-triggered CHK1-E2F1-RRM2 pathway, which resulted in a greater loss of dNTPs and profound RS leading to a synergistic anti-tumor effect. Finally, they demonstrated that the anti-rheumatoid arthritis drug aurorafin, which is known as a TrxR1 inhibitor, also exhibited a synergistic anti-tumor activity with CHK1i. Overall, this study is novel, interesting and well designed. I listed several concerns below for the authors to improve the manuscript.

Major points:

1. The synergistic effect of Trx1 knockdown/inhibition with CHK1i on RS induction and subsequent cell death is quite convincing. However, there seems contradictory in terms of the mechanistic findings. The authors demonstrated that knockdown of Trx1 induced RRM1 oxidation, which is critical for Trx1 depletion-induced RS (Figure 6A-C). On the other hand, they showed that CHK1i can abrogate Trx1 knockdown-induced RRM1 oxidation (Figure 7G-H, right panel). Thus, one may expect that CHK1 inhibition could have the capability to dampen the effect of Trx1 knockdown on RS and subsequent cell death, but this is not the case. The authors should provide a reasonable explanation to reconcile these findings.
2. Since this work aims to identify novel determinants of CHK1i sensitivity and develop effective strategies to enhance CHK1i treatments, the working model in Figure 9M seems a little weird. It seems more reasonable to draw a working model illustrating how NSCLC cells are not sensitive to CHK1i alone, and then show a new one illustrating how Trx1 or TrxR1 depletion or inhibition could synergistically interact with CHK1i to exert a more dramatic anti-tumor effects than either monotherapy alone.
3. There exists a redundancy between cytosolic Trx1 and the Grx-GSH system, and most of the results were obtained from H1299 and Calu-6 cells which have defective Grx-GSH system. Did the authors also observe the synthetic lethal effect by combinational CHK1 and TrxR1 inhibitor treatment in other NSCLC cell lines with intact Grx-GSH system?

Minor points:

1. In Figure 3J, there were six treatment groups showing the representative immunofluorescent images. However, there were 10 groups in the corresponding quantitation data as shown in Figure 3K.
2. In Figure 5D, the statistical analysis was missing.
3. In Figure 5H, the dose-dependent effects of dNTP addition on the relative CldU track

length in the shTrxR1 groups were not convincing. Similar issue for the effects of dNTP addition on the levels of p-RPA32(S4/8) in the shTrx1 groups in Figure 5I, and the effects of AUR treatment on the levels of γ H2AX in the CHK1 treatment groups in Figure 9G. Any explanation?

4. In the legends of Figure S5, the “shTrx1-1 C35S” should be “shTrx1-1 Trx1 C35S”.
5. In Figure S6, the data of dATP were missing.
6. In Figure 8B, “+1 μ M AUR” group is not indicated.
7. In Figure S3D-F, “pLKO “should be “shCON”.

Reviewer #3 (Remarks to the Author):

In present study, Zhang and colleagues identified the thioredoxin-1 (TRX1) system is a novel determining factor of Chk1 inhibitor sensitivity via shRNA based high-throughput screening. They confirmed the synergistic anti-tumor effect of combined-treatment of TRX1 inhibitor and Chk inhibitor in NSCLC cells in vitro and in vivo. Mechanistically, they found TRX1 inhibition elicits dNTPs deficiency-induced DNA replication stress, which is resulted from inactivation of RRM1-mediated ribonucleotide reductase (RNR) activity. This study provided a promising combination strategy to optimize efficacy of Chk1 inhibitor in NSCLC cancers and combination treatment of TrxR1 inhibitor auronafin with Chk1 inhibitor displays significant translation value. However, TrxR1 system has been known as an electron donor for RNR reaction in bacteria and T cells(J Biol Chem. 1964 Oct;239:3436-44.; Proc Natl Acad Sci U S A. 1976 Jul;73(7):2275-9. Free Radic Biol Med. 2010 Dec 1;49(11):1617-28.; Nat Commun. 2018 May 10;9(1):1851.), and it is well-established that triggering DNA replication stress via disruption RNR induces Chk1 inhibitor hypersensitivity. Thus, I failed to find any advanced and conceptual novelty in presented data, the present manuscript is more suitable to publish in more specialized journal.

Major points:

1. The efficacy of AUR and Chki combination could be evaluated in patients' derived NSCLC models, such as PDX or organoid to strength the clinical relevance.
2. Silence of TRX1 leads to RRM1 cysteine oxidation, whether TRX1 deficiency affects RNR enzymatic activity?
3. Whether TRX1/TRXR1 is a promising therapeutic target in cancer treatment. Although author their mRNA expressing using TCGA dataset, their protein levels were not validated in NSCLC tissues, as well as matched normal lung.
4. IHC analysis of DNA replication stress, DNA DSBs or other index could be performed in drug treated-xenografts tissues.
5. Whether TRX1 physically interacts with RRM2 or RRM1?
6. Given that TRX1 inhibition inhibits RRM1, whether targeting TRX1 could be more effective to sensitize cancer cells to RNR-targeting agents such as gemcitabine.
7. What's the effects of TRX1 inhibition on alternative RNR small subunit, p53R2?

Reviewer #1:

In this study, Prasad and colleagues investigate the mechanistic basis of thioredoxin-1/TrxR1-based modulation of Chk1i vulnerability in NSCLC lines. The authors present an interesting and physiologically valid rationale for why inhibiting the thioredoxin system enhances Chk1i-induced replication stress. This investigation goes well beyond the obvious attribution of the observed phenotype to ROS detoxification systems modulated by Trx1 function, adding significant mechanistic depth to this study. The authors propose this co-inhibition relies on dysregulation of reductive biosynthesis by ribonucleotide reductase, leading to depletion of the deoxyribonucleotide pool and replicative stress-induced genotoxic damage that is independent of ROS. The overall experimental strategy for this study is extremely well-designed, with solid consideration of the key mechanistic factors needed to support their hypothesis and with appropriate methodical rigor. In particular, there is a great deal of elegant data provided in Figs. 5 and 6 supporting involvement of replication stress, the role of a depleted nucleotide pool and RRM oxidation. However, there is some unevenness in interpretation of data and some gaps in characterization. Specific concerns are described below. Experimental and/or textual clarification provided by the authors regarding these points would significantly strengthen their innovative study.

Response: We are grateful for the positive review from the reviewers and appreciate their valuable time and thoughtful comments on the current manuscript. In addition to the 127 figures in the initial submission, we have integrated an additional 48 new figures into the revised manuscript. We hope that all the reviewers' queries have been satisfactorily addressed.

- **Fig 2:** The reduction in tumor weight and volume are very impressive. The authors are encouraged to also provide H&E images to show the overall changes in cellularity under the specific treatments. Further, characterization of tumor tissues could add mechanistic heft to these in vivo experiments and make them consistent with the extensive in vitro studies. For example, in vitro, the H1299 cells presumably undergo cell death (which would contribute to a durable treatment response as opposed to growth arrest). It would be useful to know if these significantly reduced tumor burdens are also due to induction of apoptosis. The authors should ideally provide TUNEL images or tumor western blots showing cleaved PARP or cleaved caspase levels in support of this information. Additionally, how are DNA damage markers/replication stress markers altered across the control and differentially treated tumors?

Response: Thank you for the suggestions.

We extracted total protein from the excised xenograft tumors and used it to detect markers of apoptosis and replication stress/DNA damage. The data obtained from Western blot (WB) indeed show increased PARP1 cleavage, a higher amount of DNA damage, and increased CHK1 phosphorylation, especially in the condition of a combination of Trx1 or TrxR1 knockdown with CHK1 inhibition (**Fig. S7A and S7B**). Additionally, we stained the tissues with H&E (**Fig. S4C**) and performed a TUNEL assay to assay cellular morphology and the changes in apoptosis (**Fig. 2I, J**). Indeed, an increased TUNEL intensity was observed in tissues derived from cancer cells with Trx1 or TrxR1 KD in combination with CHK1 inhibition (**Fig. 2I and 2J**).

- **Fig 2:** there is some variability in the tumor sizes although only the smallest are shown in Fig 2F. If the levels of Trx1 or TrxR1 knockdown are evaluated across the Chk1i treated tumors, do the smaller tumors have better knockdown compared to those tumors that have grown bigger? These data would directly support that co-inhibition of the Trx1 system cooperates with Chk1i-induced tumor suppression in vivo

Response: Thank you for the suggestion. We evaluated the levels of Trx1 and TrxR1 in tumor xenografts to assess the effectiveness of knockdown (KD) among experimental groups (**Fig. S7A and S7B**). The level of KD within the KD group across the samples were profoundly low, regardless of tumor size. Therefore, the levels of Trx1 or TrxR1 KD are comparable, and their potential contribution to the variation in tumor size in KD group is minimal.

- Regarding the patient-derived data analysis in Fig 1, it should be noted is that H1299 is NRAS-driven, which is not typical of human NSCLC. The Calu line is driven by KRAS G12C, the more human-representative oncogenic driver of NSCLC. KRAS-driven tumors are enriched for the thiol redox-protective interactome; however, this is not known for NRAS. To extend the disease-physiological relevance of their *in vivo* results which are in the H1299 line, authors should ideally address this by specifically evaluating public patient datasets for whether the small subset of NRAS driven LUAD also exhibit elevated Trx1, TrxR1, or if there is correlation between oncogenic NRAS and dependency on this interactome.

Response: Thank you for the suggestion.

We reanalyzed the patient-derived datasets and segregated the samples based on NRAS mutation. Only 9 out of 509 patient samples exhibit NRAS mutation. Therefore, with such a small sample size, we are unable to draw conclusions regarding the relationship between NRAS and Trx1/TrxR1 expression levels. We agree with the reviewer that investigating this synergy in a specific NRAS population would be interesting in the future.

- **Fig 3.** With the extensive degree of DNA breaks produced by this treatment, TUNEL should be supplemented by immunoblotting for molecular markers of cell death to support the cytotoxicity evoked by combinatorial treatment. If these are not seen, the S-phase arrest observed may indicate senescence-like arrest as the predominant tumor suppressor response. This distinction also impinges on my comments for Fig. 2 above. Cell death-inducing treatments in general engender a more durable anti-tumor response. If this combination is indeed evoking cell death, then this point could be reliably made in the discussion.

Response: We conducted the TUNEL assay in xenograft tumor tissue (**Fig.2I, J**), supplemented with Western blotting (WB) to detect cleaved PARP1 and the DNA damage marker γ H2AX, to confirm apoptosis *in vivo* in the group with a combination of Trx1 or TrxR1 knockdown and CHK1 inhibition (**Fig. S7A and S7B**). Additionally, we performed the apoptosis assay *in vitro* using Annexin V and propidium iodide (PI) staining in non-small cell lung cancer (NSCLC) cells treated with CHK1i followed by Trx1 or TrxR1 knockdown (**Fig. S3A and S3B**). Furthermore, we also detected cleaved PARP1 in H1299 cells treated with CHK1i followed by Trx1 or TrxR1 KD (**Fig. S3C**). Finally, we also conducted the TUNEL assay in H1299 xenograft as well as in a patient-derived xenograft (PDX) model to determine the effect of the combination of the TrxR inhibitor AUR and CHK1i-induced cell death (**Fig. S13C, S13D, S13E, S14B, S14C**). The results from all the additional apoptosis assays using different models and techniques suggest the same conclusion; namely, that the combination of Trx1/TrxR1 inhibition through knockdown or pharmaceutical inhibitor, when combined with CHK1 inhibition, leads to cell death.

- Are these cells WT for p53 or p16? If so, these markers should also be evaluated as these factors will impinge on how pervasively these treatments can be translated to patients. At the very least, a note should be included in Discussion regarding the dependency of these treatments on key tumor-suppressor pathways, based on their status in the cell lines used in this study.

Response: H1299 and Calu-6 cells harbor *TP53* mutations. We have discussed the potential impact of these types of mutation in the revised manuscript (Discussion section in the second paragraph, on page 19).

- **Fig 4:** The DCFDA flow profiles should be quantified, with appropriate statistics added. More importantly, the authors should be aware that demonstrating NAc-mediated rescue of DCFDA-detected ROS levels is no longer considered best-practice to indicate ROS-mediated phenomenon. NAc decreased signal by reducing the fluorophore itself rather than detoxifying cellular ROS (which is chemically not possible for NAc). Thus, the lack of effect of NAc on the other cellular outcomes from Trx1 or TrxR1 inhibition does not definitively rule out ROS involvement by virtue of its effect on the DCFDA profiles. In this regard, the authors should use a direct ROS scavenger to make their point and/or culture cells at lower ambient oxygen. The striking ROS (total peroxide) induction observed upon both TRX1 and TRXR1 depletion, coupled with the observed DNA breaks, almost certainly suggest some ROS involvement. These peroxides are labile and easily interconverted into other species such as hydroxyl radicals which can damage DNA and the nucleotide pool. Thus, the observed S-phase arrest, cytotoxicity and genotoxic damage is very likely the aggregate result of a depleted nucleotide pool and direct generation of breaks/nicks, which would account for the punctate and persistent DSB foci. This would not detract from the conclusions made by the authors and could add additional mechanistic nuance should other methods of ROS rescue not provide congruent results as NAc.

- To fully evaluate the role of ROS (or lack thereof) in the observed phenomena, the authors are encouraged to evaluate 8-oxoguanine and/or lipid peroxide levels, as these may be unappreciated contributors to the observed cytotoxicity of the dual treatment. At the very least, these parameters could help rule out or define the role played by ROS in the dual treatment settings.

Response: Thank you for the thoughtful suggestions.

N-acetylcysteine (NAC) is commonly used as a ROS scavenger. Addressing the reviewer's concern about NAC's potential to quench the DCFDA fluorophore itself rather than ROS, we followed the reviewer's suggestion and introduced an additional ROS scavenger, α -tocopherol (Vitamin E), to neutralize the ROS induced by Trx1/TrxR1 inhibition in our experimental setup. Cells were treated with α -tocopherol, and we evaluated Trx1/TrxR1 KD-induced replication stress (RS) via Western blotting (WB). We observed no significant change in Trx1/TrxR1 KD-induced RS with α -tocopherol treatment (**Fig. 4D**), consistent with our findings using NAC (**Fig. 4C**). Therefore, ROS is not a major contributor to the RS induced by Trx1 or TrxR1 inhibition and the synergy between Trx system interruption and CHK1.

In line with the suggestion, we further confirmed the rise in ROS following Trx1/TrxR1 KD by assessing lipid peroxidation in Trx1/TrxR1 KD cells, with or without NAC or α -tocopherol. The heightened ROS levels due to Trx1/TrxR1 KD led to increased lipid peroxidation, and treatment with either NAC or α -tocopherol significantly reduced lipid peroxidation (**Fig. 4B and S8B**).

- Auranofin is used in these studies to chemically inhibit TrxR1 in addition to genetic knockdown, which is appropriate and rigorous. Although this is intended to support the lack of Trx1 site reductive regeneration by TrxR1, which is critical for RRM function, TrxR1 inhibition can affect cancer cell proliferation and survival through Trx1-independent mechanisms (i.e., by inhibition of protein tyrosine phosphatase 1B function). As Trx1 directly influences RRM function, the authors are encouraged to repeat the in vitro experiments with the Trx1 specific inhibitor, PX-12. The

outcomes would likely be informative either way about the mechanisms by which the thioredoxin interactome synergizes with Chk1 inhibition. If the Px-12 results do not directly recapitulate auranofin, there may be a signaling component (through Ask1 or ATM/ATR downstream signaling) that controls the response to Chk1 inhibition, in addition to replicative stress.

Response: As recommended, we utilized the Trx-specific inhibitor PX-12 and observed that PX-12 effectively inhibited the proliferation of NSCLC cell lines in a synthetic lethal manner when combined with the CHK1 inhibitor (**Fig. S16A**). The combined treatment of PX-12 and CHK1 inhibitor resulted in an increased level of cleaved PARP1 (**Fig. S16B**). Additionally, we discovered that PX-12 treatment led to an increase in the oxidation of RRM1 and a decrease in dNTPs (**Fig. S16C**). Consequently, our data, obtained with the Trx inhibitor PX-12, demonstrates a similar effect in inducing a synthetic lethal interaction with CHK1i in the treatment of NSCLC, which is similar to what we found in the cells treated with combined TrxR inhibitor Auranofin and CHK1i.

- Textual point – please label immunoblots with molecular weight markers for the various signals provided.

Response: Molecular weight markers are indicated in the revised manuscript.

Reviewer #2 (Remarks to the Author):CHK1 inhibition (CHK1i) adjuvant therapy has shown promising efficacy in preclinical models but only minimal efficacy with substantial toxicity in clinical trials. To identify novel strategies to enhance CHK1iefficacy, Prasad et al. have identified the master redox regulator Trx1 as a new determinant of CHK1isensitivity through a high throughput screening. The authors showed that Trx1 knockdown/inhibition combined with CHK1i had a synergistic effect potentially via triggering synthetic lethality in NSCLC. They further found that Trx1 knockdown in combination with CHK1i promoted profound replication stress (RS)and cell death. Mechanistically, they found that Trx1 knockdown/inhibition abrogated dNTP production and impaired replication fork elongation to induce RS mainly through increasing the oxidation of ribonucleotide reductase (RNR) subunit RRM1, but not ROS. Moreover, CHK1i treatment disrupted theTrx1 depletion-triggered CHK1-E2F1-RRM2 pathway, which resulted in a greater loss of dNTPs and profound RS leading to a synergistic anti-tumor effect. Finally, they demonstrated that the anti-rheumatoid arthritis drug auranofin, which is known as a TrxR1 inhibitor, also exhibited a synergistic anti-tumor activity with CHK1i. Overall, this study is novel, interesting and well designed. I listed several concerns below for the authors to improve the manuscript.

Response:

We are grateful for the positive review from the reviewers and appreciate their valuable time and thoughtful comments on the current manuscript. In addition to the 127 figures in the initial submission, we have integrated an additional 48 new figures into the revised manuscript. We hope that all the reviewers' queries have been addressed and satisfied.

Major points:

1. The synergistic effect of Trx1 knockdown/inhibition with CHK1i on RS induction and subsequent cell death is quite convincing. However, there seems contradictory in terms of the mechanistic findings. The authors demonstrated that knockdown of Trx1 induced RRM1 oxidation, which is critical for Trx1depletion-induced RS (Figure 6A-C). On the other hand, they showed that CHK1i can abrogate Trx1knockdown-induced RRM1 oxidation (Figure 7G-H, right panel). Thus, one may expect that CHK1inhibition could have the capability to dampen the effect of Trx1 knockdown on RS and subsequent cell death, but this is not the case. The authors should provide a reasonable explanation to reconcile these findings.

Response: Thank you very much for comment.

Our hypothesis is that the disruption of the Trx system induces oxidation of RRM1, a crucial step for the Trx system interruption-induced replication stress (RS). This occurs by impairing dNTP production due to decreased RNR activity. When CHK1 activity is inhibited in these cells, CHK1 inhibition can further hinder RNR activity by affecting RRM2 through an E2F1-mediated mechanism. When Trx1 inhibition is combined with CHK1i, a more severe reduction in RNR activity is observed, resulting in a more profound impact on dNTP levels and tumor growth suppression. We have clarified this point in several places (Discussion section in the first paragraph, on page 18) and also revised our model accordingly (**Fig. 9M**).

An additional noteworthy finding in our study is that CHK1 inhibition leads to a decrease in Trx1 KD-induced RRM1 oxidation. This result aligns perfectly with the well-established notion in yeast and bacteria that the small subunit R2 (equivalent to human RRM2) contains a diiron-tyrosyl radical cofactor and operates as a radical chain initiator. This process generates a thiyl radical on

cysteine Cys439 in the active site of R1 (equivalent to human RRM1) via a long-range radical transfer pathway. RRM2 is a radical chain initiator. Thus, RRM2 is required for the initial RRM1 oxidation. When Chk1 is inhibited, the E2F1-RRM2 mediated pathway is blocked, leading to a decrease in Trx inhibition-induced RRM1 oxidation, although it remains higher than in control cells. This, in turn, results in the decrease of Trx1 knockdown induced RRM1 oxidation. In support of this hypothesis, hydroxyurea, a well-known RRM2 inhibitor demonstrated a similar impact on Trx1 KD-induced RRM1 oxidation (**Fig. 7G**)

2. Since this work aims to identify novel determinants of CHK1i sensitivity and develop effective strategies to enhance CHK1i treatments, the working model in Figure 9M seems a little weird. It seems more reasonable to draw a working model illustrating how NSCLC cells are not sensitive to CHK1i alone, and then show a new one illustrating how Trx1 or TrxR1 depletion or inhibition could synergistically interact with CHK1i to exert a more dramatic anti-tumor effects than either monotherapy alone. There exists a redundancy between cytosolic Trx1 and the Grx-GSH system, and most of the results were obtained from H1299 and Calu-6 cells which have defective Grx-GSH system. Did the authors also observe the synthetic lethal effect by combinational CHK1 and TrxR1 inhibitor treatment in other NSCLC cell lines with intact Grx-GSH system?

Response:

We agree that our goal is to identify new factors influencing CHK1 inhibitor sensitivity and develop effective strategies to improve CHK1 inhibitor treatments. However, a significant aspect of this study is to uncover the regulatory role of the Trx system in the function of RNR and dNTP pool in mammalian cells—a topic that has been overlooked in the field. Our discovery of how the Trx system regulates RNR function serves as the basis for the synergy between Trx system inhibition and CHK1 inhibition. This is why a section of our model demonstrates that the Trx system regulates RNR (RRM1). However, we indeed incorporated the suggestions from the reviewer into this revised model by including RRM2, making it clearer and more complete. Thank you!

Minor points:

1. In Figure 3J, there were six treatment groups showing the representative immunofluorescent images. However, there were 10 groups in the corresponding quantitation data as shown in Figure 3K.

Response: This is because two shRNAs were employed for each gene. However, due to space limitations, only the image from one of the two shRNAs is presented. The experiments were conducted across all 10 indicated groups, and **Figure 3J** represents an image of shCON and one shRNA for Trx1 and TrxR1.

2. In Figure 5D, the statistical analysis was missing.

Response: Statistical analysis is now incorporated in Figure 5D.

3. In Figure 5H, the dose-dependent effects of dNTP addition on the relative CldU track length in the shTrxR1 groups were not convincing. Similar issue for the effects of dNTP addition on the

levels of p-RPA32(S4/8) in the shTrx1 groups in Figure 5I, and the effects of AUR treatment on the levels of γ H2AX in the CHK1 treatment groups in Figure 9G. Any explanation?

Response:

We attempted various concentrations of the dNTP pool and different treatment durations to alleviate the RS induced by Trx1/TrxR1 KD. Due to the dynamic nature of replication in actively proliferating cells and the variable concentrations of the four dNTPs, it is nearly impossible to fully restore the Trx1/TrxR1 KD-induced RS by reconstituting all four dNTPs to biological levels. We used equimolar concentrations of all four dNTPs to rescue the RS phenotype. But it's important to note that cells do not naturally produce all dNTPs in equimolar concentration even during normal replication. Consequently, we do not anticipate a complete RS rescue under our experimental conditions. Nevertheless, there is a discernible trend showing a decrease in p-CHK1 and p-RPA32 in KD groups after the addition of dNTPs at concentrations of 50 and 100 μ M (**Figure 5I**), supported by the fiber assay in **Figure 5H**. Additionally, we observed a dose-dependent increase in γ H2AX in cells treated with combined AUR and CHK1 inhibitor in **Figure 9G**.

4. In the legends of Figure S5, the “shTrx1-1 C35S” should be “shTrx1-1 Trx1 C35S”.

Response: In the revision Fig. S5 has been changed to Fig. S10. The changes are incorporated at the indicated place.

5. In Figure S6, the data of dATP were missing.

Response: Data for the dATP was already provided in Figure 7D in the initial submission. Therefore, the data of Fig. S6 in the initial submission did not show it again. In the resubmission, Figure 7D remain unchanged, and Figure S6 has been changed to Fig. S11.

6. In Figure 8B, “+1 μ M AUR” group is not indicated.

Response: This recommendation is incorporated into Figure 8B of the revision.

7. In Figure S3D-F, “pLKO” should be “shCON”.

Response: In the revision Figure S3D-F has been changed to Fig. S5D-F. The recommendation is incorporated into Figure S5D-F.

Reviewer #3 (Remarks to the Author): In present study, Zhang and colleagues identified the thioredoxin-1 (TRX1) system is a novel determining factor of Chk1 inhibitor sensitivity via shRNA based high-throughput screening. They confirmed the synergistic anti-tumor effect of combined treatment of TRX1 inhibitor and Chk inhibitor in NSCLC cells in vitro and in vivo. Mechanistically, they found TRX1 inhibition elicits dNTPs deficiency-induced DNA replication stress, which is resulted from inactivation of RRM1-mediated ribonucleotide reductase (RNR) activity. This study provided a promising combination strategy to optimize efficacy of Chk1 inhibitor in NSCLC cancers and combination treatment of TrxR1 inhibitor auranofin with Chk1 inhibitor displays significant translation value. However, TrxR1 system has been known as an electron donor for RNR reaction in bacteria and T cells (J Biol Chem. 1964 Oct;239:3436-44.; Proc Natl Acad Sci U S A. 1976 Jul;73(7):2275-9. Free Radic Biol Med. 2010 Dec 1;49(11):1617-28.; Nat Commun. 2018 May 10;9(1):1851.), and it is well-established that triggering DNA replication stress via disruption RNR induces Chk1 inhibitor hypersensitivity. Thus, I failed to find any advanced and conceptual novelty in presented data, the present manuscript is more suitable to publish in more specialized journal.

Response: We are grateful for the positive review from the reviewers and appreciate their valuable time and thoughtful comments on the current manuscript. In addition to the 127 figures in the initial submission, we have integrated an additional 48 new figures into the revised manuscript. We hope that all the reviewers' queries have been addressed and satisfied.

It is noteworthy that, while the association between RNR activity and CHK1 inhibition sensitivity has been suggested in the past, the synergy between Trx system inhibition and CHKi in treating NSCLC has not been reported. Most importantly, a significant aspect of our study is to reveal the regulatory role of the Trx system in the function of RNR and its relation to the replication stress in mammalian cells—a topic that has been overlooked in the field. As we discussed in the previous version of the manuscript, the regulation of RNR by the Trx system is based on results from *E. coli* and/or yeast. The studies with mammalian cells are limited to the biochemical assays determining RNR activity regeneration by the Trx system, while RNR redox status *in vivo* has not been investigated. Therefore, the contribution of RRM1 oxidation to replication stress and its role in cancer therapy have not been previously suggested, which is one of the key findings of our study. Furthermore, our study establishes the molecular basis for the synergistic interaction between the Trx system and CHK1 inhibition. Lastly, the involvement of both RRM1 oxidation and RRM2 downregulation in the synergistic interaction between the Trx system and CHK1 is novel.

We apologize for not being able to discuss this paper (Nat Commun. 2018) clearly in the previous submission. Although it was suggested that the Trx system is crucial for supporting DNA synthesis during T-cell metabolic reprogramming this study did not explore the regulatory role of the Trx system in RNR (RRM1). In contrast, this study utilized indirect evidence from non-targeted metabolomics analysis and pathway enrichment analysis under stimulated conditions to predict the potential role of the Trx system in metabolic reprogramming. The study used the accumulation of nucleotides containing ribose sugar as a metric to suggest that inhibiting the Trx system leads to a defective final reduction of RNA into DNA building blocks during DNA biosynthesis. Consequently, the conclusion was drawn that the Trx system in T cells regulates the reduction of RNA into DNA building blocks at the last step of DNA biosynthesis. However, it's important to note that this paper is not directly relevant to our study, as it did not include functional regulation of RNR. Additionally, there is no evidence in the paper demonstrating that the Trx system regulates RRM1 oxidation through an interrupted long electron transfer chain.

Major points:

1. The efficacy of AUR and Chki combination could be evaluated in patients' derived NSCLC models, such as PDX or organoid to strength the clinical relevance.

Response: Thank you for the suggestion. To further assess the combined efficacy of CHK1i and Auranofin (AUR) in treating NSCLC, we employed a Patient-Derived Xenograft (PDX) model. The results from the PDX model (PDX72) of NSCLC have been incorporated into the revised manuscript. We observed a significant reduction in tumor growth in PDX72 when treated with the combination of AUR and CHK1i compared to treatment with each compound alone (**Fig. 8J-8M**). The combined treatment also led to a noteworthy increase in the survival rate of mice (**Fig. 8K**). Tunnel assays suggested a significant apoptosis in combinational group, compared to each drug alone (**Fig. S14B and S14C**).

Additionally, we generated organoids from the same PDX tissue sample. Similarly, the rate of organoid formation significantly decreased when CHK1i was introduced along with the TrxR inhibitor Auranofin or the Trx1 inhibitor PX-12 (**Fig. 8P and 8Q**). We evaluated cell death using Propidium Iodide (PI) staining in the organoids and observed a higher rate of cell death following the combined treatment (**Fig. 8N and 8O**). Additionally total protein isolated from organoid treated with combined inhibitors shows increased level of PAPER1 cleavage (**Fig. S14E**). Therefore, the synergistic interaction between Trx1 or TrxR1 inhibition and CHK1i was also validated in both the PDX and organoid models.

2. Silence of TRX1 leads to RRM1 cysteine oxidation, whether TRX1 deficiency affects RNR enzymatic activity?

Response: It has been demonstrated that Trx1 affects the RRM1 regeneration activity using biochemistry assay (J Biol Chem. 2017 Jun 2;292(22):9136-9149).

3. Whether TRX1/TRXR1 is a promising therapeutic target in cancer treatment. Although author their mRNA expressing using TCGA dataset, their protein levels were not validated in NSCLC tissues, as well as matched normal lung.

Response: Thank you for the suggestion. We utilized a commercially available cohort of NSCLC tissue microarray (TMA) and corresponding adjacent normal tissue to evaluate the expression of Trx1 and TrxR1 via IHC. Our findings revealed elevated levels of Trx1 and TrxR1 expression in NSCLC tissue compared to the matched normal tissue (**Fig.1D-F**). Therefore, the IHC staining of the TMA aligns with the results obtained from the analysis of The TCGA data, indicating a higher level of Trx1 and TrxR1 in NSCLC in comparison to normal tissue.

4. IHC analysis of DNA replication stress, DNA DSBs or other index could be performed in drug treated-xenografts tissues.

Response: Thank you for the suggestion. We conducted WB to assess RS in xenograft tissue with KD and drug treatments (**Fig. S7A and S7B, S13E and S13F**). Additionally, we performed TUNEL assays in the xenograft model and in PDX model to evaluate cell death and apoptosis (**Fig. S13 C-F**). Finally, we carried out TUNEL assay using the Patient-Derived Xenograft (PDX) model to validate the effects of drug treatment on apoptosis (**S14B-S14C**).

5. Whether TRX1 physically interacts with RRM2 or RRM1?

Response: Yes. the physical interaction between Trx1 and RRM1 has been demonstrated through their cysteine moieties. The physical interaction between RRM1 and Trx1 was previously demonstrated (PMID: 28411237).

6. Given that TRX1 inhibition inhibits RRM1, whether targeting TRX1 could be more effective to sensitize cancer cells to RNR-targeting agents such as gemcitabine.

Response: Given that hydroxyurea is a well-known chemotherapy drug targeting RNR activity, we now provide evidence showing that hydroxyurea treatment leads to a RRM2-mediated long electronic chain transfer impairment like how CHK1i can block the E2F1-RRM2 pathway (**Fig.6I, Fig.7G, H**). Therefore, hydroxyurea is a better RNR targeting agent to test our hypothesis. We performed a cytotoxic assay and found that Trx1 or TrxR1 KD cells treated with hydroxyurea show markedly greater sensitivity than KD alone (**Fig. 1** in this rebuttal letter – see below).

7. What's the effects of TRX1 inhibition on alternative RNR small subunit, p53R2?

Response: We measured the level of p53R2 in Trx1/TrxR1 KD H1299 cells and observed no significant difference in the protein expression of p53R2 in KD group compared to the control group (**Fig. 2** in this rebuttal letter – see below).

[Redacted]

Figure 1. Cytotoxic effect of Hydroxyurea (HU) on Trx1 or TrxR KD cells.

[Redacted]

Figure 2. No significant difference in the protein expression of p53R2 among the cells with or without Trx1 or TrxR1 KD.

REVIEWER COMMENTS

Reviewer #1 (Remarks to the Author):

Overall, the authors are to be congratulated on their thorough and carefully considered revisions. They have added multiple pieces of data that strengthen their core conclusions. I have some follow up comments regarding relatively minor textual changes to improve the accuracy and rigor of their data:

1. Fig S4C. The H&E images are simply provided without linking to any conclusion in the manuscript. The rationale for asking for H&E is to have images that support their conclusion that their treatment group shows the changes in cellularity consistent with their assertion that Chk1i in conjunction with TrxR or Trx inhibition induces cell death. This is not clear from the current images which should be in higher magnification and possibly higher resolution (unclear if the low-res is just in the reviewer pdf). Arrows indicate areas of low cellularity or nuclear features consistent with cell death.
2. In Fig S7, it is unclear if the 4 tumors for which Trx or TrxR immunoblotting are different or similar in size, so please add the tumor volumes next to the tumor numbers. If the knockdown is not correlated to tumor volume, then there may be some additional factor(s) which affect the anti-tumor response (this is not unusual in these types of studies and may require additional markers of efficacy that need to be evaluated; this fact should be noted in the manuscript as needed).
3. Fig S8B – please add LAA to the figure labels, unclear which groups were treated from the legend. I could not find a description in the text for any LAA experiments.
4. More importantly, to my knowledge, tocopherol does not scavenge the species measured by DCFDA (hydrogen peroxides). It can however scavenge lipid peroxy radicals. Therefore the section starting on line 245 needs inclusion of references to the functions of the scavengers used in these studies to clarify the rationale for using them. The authors make it clear that ROS involvement is not a key factor in the observed replication stress, despite the changes observed in oxidant species. Nevertheless, the requested clarification is important for accuracy.
5. Lines 639 -643 – the current discussion of p53 status is somewhat perfunctory and should be developed more specifically. At the very least, discuss HOW p53 status is reported to affect checkpoint inhibitor efficacy and add a reference or two. As their lines harbor p53 mutations and still respond to the Chk1i-based combinations, the potential for therapeutic translational in NSCLC is increased as > 50% of these tumors lack functional p53. A sentence or two of how apoptosis is being induced effectively in the absence of wt p53 should be added as well.

Reviewer #2 (Remarks to the Author):

There are some sloppiness here. For the major point 3, the authors didn't give any response. Moreover, for the minor point 6, the authors mentioned that "This recommendation is incorporated into Figure 8B of the revision". However, the Figure 8B is missing in the resubmission.

Reviewer #3 (Remarks to the Author):

Authors appropriately addressed my concerns and I have no further questions

In response to the reviewer's feedback, we have made minor textual adjustments as requested. A point-by-point response is provided in blue in the second revision.

Reviewer #1 (Remarks to the Author):

Overall, the authors are to be congratulated on their thorough and carefully considered revisions. They have added multiple pieces of data that strengthen their core conclusions. I have some follow up comments regarding relatively minor textual changes to improve the accuracy and rigor of their data:

Response: We thank the reviewer for his/her positive comment on the revision.

1. Fig S4C. The H&E images are simply provided without linking to any conclusion in the manuscript. The rationale for asking for H&E is to have images that support their conclusion that their treatment group shows the changes in cellularity consistent with their assertion that Chk1i in conjunction with TrxR or Trx inhibition induces cell death. This is not clear from the current images which should be in higher magnification and possibly higher resolution (unclear if the low-res is just in the reviewer pdf). Arrows indicate areas of low cellularity or nuclear features consistent with cell death.

Response: The H&E images of higher magnification are incorporated in the manuscript with indicated changes in cellularity (Fig. S4C and Fig. S14A).

2. In Fig S7, it is unclear if the 4 tumors for which Trx or TrxR immunoblotting are different or similar in size, so please add the tumor volumes next to the tumor numbers. If the knockdown is not correlated to tumor volume, then there may be some additional factor(s) which affect the anti-tumor response (this is not unusual in these types of studies and may require additional markers of efficacy that need to be evaluated; this fact should be noted in the manuscript as needed).

Response: Thank you for your suggestion. However, the tumors for immunoblotting were randomly selected from experimental groups. As a result, we were unable to trace the tumor size for each individual paired sample. We appreciate this consideration and will include it in future studies.

3. Fig S8B – please add LAA to the figure labels, unclear which groups were treated from the legend. I could not find a description in the text for any LAA experiments.

Response: The LAA description has been added to the figure legend for Fig S8B in the first revision. The description of the method was provided in Supplementary Information on page 27 as an addition Methods section.

4. More importantly, to my knowledge, tocopherol does not scavenge the species measured by DCFDA (hydrogen peroxides). It can however scavenge lipid peroxy radicals. Therefore, the section starting on line 245 needs inclusion of references to the functions of the scavengers used in these studies to clarify the rationale for using them. The authors make it clear that ROS

involvement is not a key factor in the observed replication stress, despite the changes observed in oxidant species. Nevertheless, the requested clarification is important for accuracy.

Response: It has been demonstrated that peroxy radicals can cause oxidative DNA damage and lead to lipid peroxidation-induced alterations in DNA¹. Other free radicals, such as singlet oxygen species (¹O₂), can also trigger oxidative damage to DNA and oxidize the guanine base to 8-oxo-guanine, resulting in a cellular DDR response². Alpha-tocopherol can significantly scavenge both peroxy and singlet oxygen species^{3,4}. Therefore, in addition to NAC, we also used alpha-tocopherol as a second scavenger to remove the DNA damage-inducing reactive oxygen species (ROS) produced in cells upon inhibition of the Trx system.

See page 8 for detailed explanation in the main manuscript.

5. Lines 639 -643 – the current discussion of p53 status is somewhat perfunctory and should be developed more specifically. At the very least, discuss HOW p53 status is reported to affect checkpoint inhibitor efficacy and add a reference or two. As their lines harbor p53 mutations and still respond to the Chk1i-based combinations, the potential for therapeutic translational in NSCLC is increased as > 50% of these tumors lack functional p53. A sentence or two of how apoptosis is being induced effectively in the absence of wt p53 should be added as well.

Response: p53 is important for the G1/S checkpoint. Disruption of G1/S checkpoint control due to loss of p53 leaves cells reliant on cell cycle G2/M arrest for DNA repair when the cells are challenged with DNA damaging agents. CHK1 phosphorylates and inhibits its substrates, the phosphatases CDC25C and CDC25A, leading to arrest at the G2/M checkpoint. Therefore, p53-deficient cells are normally more sensitive to CHK1 inhibitor-associated cancer therapy.

See the first paragraph on page 20 for detailed discussion in the main manuscript.

Reviewer #2 (Remarks to the Author):

There are some sloppiness here. For the major point 3, the authors didn't give any response. Moreover, for the minor point 6, the authors mentioned that "This recommendation is incorporated into Figure 8B of the revision". However, the Figure 8B is missing in the resubmission.

Response: We want to clarify that we have addressed point 3 in the previous revision of article and in the cover letter. However, we do apologize for our oversight where the detailed response to point 3 in the first rebuttal letter was accidentally lost during the editing process although we have already addressed this question in the cover letter and article in the first revision.

The concern raised by reviewer 2 regarding the synthetic lethal effect achieved through the combinatorial treatment with a CHK1 and TrxR inhibitor in other NSCLC cell lines with an intact Grx-GSH system (point 3) has already been addressed in the first revision. We found that Trx1 or TrxR1 KD had no obvious impact on RS (Fig. S6A-D) and cytotoxicity (Fig. S6E) in the A549 cells that have an intact GSH system. The corresponding results were presented in supplementary Fig. S6A-E in the first revision. We have kept the figure unchanged in the second revision.

We apologize for the confusion. The legend for "+1 μ M AUR" was incorporated in the first revision. The previous panel Figure 8B has been moved to the supplementary figure S12B in the first revision. Thus, the current Figure 8B is a different figure.

- 1 Lim, P. *et al.* Peroxyl radical mediated oxidative DNA base damage: implications for lipid peroxidation induced mutagenesis. *Biochemistry* **43**, 15339-15348, doi:10.1021/bi048276x (2004).
- 2 Ravanat, J. L. & Dumont, E. Reactivity of Singlet Oxygen with DNA, an Update. *Photochem Photobiol* **98**, 564-571, doi:10.1111/php.13581 (2022).
- 3 Niki, E. Role of vitamin E as a lipid-soluble peroxyl radical scavenger: in vitro and in vivo evidence. *Free Radic Biol Med* **66**, 3-12, doi:10.1016/j.freeradbiomed.2013.03.022 (2014).
- 4 Tasaka, T., Matsumoto, T., Nagashima, U. & Nagaoka, S.-i. Potential energy curve for singlet-oxygen quenching reaction by vitamin E. *Journal of Photochemistry and Photobiology A: Chemistry* **442**, 114749, doi:https://doi.org/10.1016/j.jphotochem.2023.114749 (2023).

REVIEWERS' COMMENTS

Reviewer #2 (Remarks to the Author):

The authors have answered my concerns and the manuscript is ready for the publication.